# KDEL receptor regulates secretion by lysosome relocation- and autophagy-dependent modulation of lipid-droplet turnover

Diego Tapia[1,2], Tomás Jiménez[1,2], Constanza Zamora[1,2], Javier Espinoza[2], Riccardo Rizzo[3], Alexis González-Cárdenas[4], Danitza Fuentes[5], Sergio Hernández[1], Viviana A. Cavieres[1,4], Andrea Soza[1,6], Fanny Guzmán[7,8], Gloria Arriagada [2], María Isabel Yuseff [5], Gonzalo A. Mardones[1,4], Patricia V. Burgos[1,6], Alberto Luini[3], Alfonso González[1,6,9] & Jorge Cancino[1]

Inter-organelle signalling has essential roles in cell physiology encompassing cell metabolism, aging and temporal adaptation to external and internal perturbations. How such signalling coordinates different organelle functions within adaptive responses remains unknown. Membrane traffic is a fundamental process in which membrane fluxes need to be sensed for the adjustment of cellular requirements and homeostasis. Studying endoplasmic reticulum-to-Golgi trafficking, we found that Golgi-based, KDEL receptor-dependent signalling promotes lysosome repositioning to the perinuclear area, involving a complex process intertwined to autophagy, lipid-droplet turnover and Golgi-mediated secretion that engages the microtubule motor protein dynein-LRB1 and the autophagy cargo receptor p62/SQSTM1. This process, here named 'traffic-induced degradation response for secretion' (TIDeRS) discloses a cellular mechanism by which nutrient and membrane sensing machineries cooperate to sustain Golgi-dependent protein secretion.

[1] Centro de Biología Celular y Biomedicina (CEBICEM), Facultad de Medicina y Ciencia, Universidad San Sebastián, Lota 2465, 7510157 Santiago, Chile. [2] Departamento de Ciencias Biologicas, Facultad de Ciencias de la Vida, Universidad Andrés Bello, Quillota 980, Viña del Mar 2520000, Chile. [3] Consiglio Nazionale delle Ricerche (CNR), Istituto di Biochimica delle Proteine (IBP), Via Pietro Castellino 111, 80131 Napoli, Italy. [4] Instituto de Fisiología, Facultad de Medicina, Universidad Austral de Chile, 5110566 Valdivia, Chile. [5] Departamento de Biología Celular y Molecular, Facultad de Ciencias Biológicas, Pontificia Universidad Católica de Chile, 8331150 Santiago, Chile. [6] Centro de Envejecimiento y Regeneración (CARE-UC), Facultad de Ciencias Biológicas, Pontificia Universidad Católica de Chile, 8330023 Santiago, Chile. [7] Fraunhofer Chile Research, 7550296 Santiago, Chile. [8] Núcleo Biotecnología Curauma, Pontificia Universidad Católica de Valparaíso, 2373223 Valparaíso, Chile. [9] Fundación Ciencia y Vida, Zañartu 1482, 7750000 Santiago, Chile. Correspondence and requests for materials should be addressed to J.C. (email: jorge.cancino@uss.cl)

A defining feature of eukaryotic cells is the compartmentalization of precise and specific functions into membrane-limited organelles. Although often conceived as separate entities, organelles are neither functionally nor structurally isolated. The endoplasmic reticulum (ER), mitochondria, nucleus, plasma membrane (PM) and the Golgi complex physically interact during dynamic communicative processes, yet preserving their compartmentalization[1,2]. These inter-organelle interactions accomplish essential tasks in many physiological processes, such as ageing, cell metabolism and signalling, and the spatiotemporal adaptation to stress[3–6]. The distribution of organelles also rapidly becomes asymmetric under several conditions. For example: developing neurons reposition their centrosome and Golgi complex towards sites of neurite outgrowth;[7] migrating cells establish rearward positioning of the nucleus as they move following attractant cues;[8] cells of the immune system polarize secretory vesicles towards immune synapses;[8,9] nutrient starvation leads to reposition of lysosomes for autophagy[10]. Extensive inter-organelle communication-dependent processes and cross-regulation occurs through contact sites without membrane fusion[11–15]. To date, the most characterized of these processes have been $Ca^{2+}$ homeostasis, lipid trafficking and autophagosome formation[10,16–18]. However, our understanding of how physiological perturbations elicit coordinated organelle positioning with functional consequences is far from complete.

During secretion, trafficking cargo proteins are first transported from the ER to the Golgi complex and then from the trans-Golgi network to the cell surface. We recently described the molecular architecture of a Golgi-based control system that regulates membrane trafficking[19]. This little understood control system is based on the recently discovered function of the KDEL receptor (KDELR) as a Golgi-localized G protein-coupled receptor (GPCR)[20,21]. We have previously established that KDELR becomes activated by KDEL-bearing chaperones during ER-to-Golgi membrane trafficking, and independently of the kind of cargo and cell type[19,20,22]. The KDELR acts as a sensor that modulates the membrane trafficking machinery, and exerts transcriptional control on secretion-related and non-related organelles[19,23]. An attractive possibility remaining to be explored is that, as a membrane trafficking-stimulated GPCR, KDELR might coordinate inter-organelle cooperation to sustain protein secretion.

Because lysosomes are secretion-related organelles linked to both the exocytic and endocytic routes, we decided to analyse their role during biosynthetic secretion. Although lysosomes were initially considered simply cellular 'incinerators' that degrade and recycle cellular waste[24], this over-simplified view has deeply evolved. Lysosomes are now recognized as organelles crucially involved in cell signalling and energy metabolism, key regulators of cell homeostasis[24–26]. As such, cell homeostasis equally depends on the fusion of lysosomes and autophagosomes for the completion of autophagy, a cellular adaptive 'self-eating' process[10]. Here, we show that ER-to-Golgi, protein trafficking-mediated activation of the KDELR signalling pathway induces relocation of lysosomes to the perinuclear region of the cell. We provide a detailed molecular characterization of this process that we named 'traffic-induced degradation response for secretion' (TIDeRS). TIDeRS engages at least three functional cellular modules: the machinery for membrane transport along the secretory route, the autophagy machinery and the cytoskeleton, involving microtubule molecular motors. Moreover, maintenance of Golgi-to-plasma-membrane overload of protein transport requires relocation of lysosomes, as well as autophagy-dependent lipid-droplet turnover. Thus, TIDeRS reveals a novel and unsuspected function of lysosomes in the biosynthetic secretory route, at the Golgi level.

## Results

**ER-to-Golgi trafficking induces lysosome repositioning**. In experiments designed to visualize the synchronized transport from the ER of a newly synthesized lysosomal protein (LAMP1-GFP (green fluorescent protein)), we observed that lysosomes, which initially were located throughout the cytoplasm (Fig. 1a, ER), moved towards the Golgi complex at about the same time the lysosomal protein reached this organelle (Fig. 1a, Golgi). Exit from the Golgi complex of this lysosomal protein resulted in its transport to lysosomes, which again relocated to an apparent initial cytoplasmically spread distribution (Fig. 1a, post-Golgi). A quantitative analysis showed that the proportion of cells with lysosome repositioning to the perinuclear region occurred transiently when cargo reached the Golgi complex (Fig. 1a, bar graph). We also tracked the synchronized release from the ER of an exocytic transport reporter, the human growth hormone fused to the polymerization/depolymerization FM domain (hGH-GFP-FM)[27,28]. Likewise, while this cargo reached the Golgi complex, lysosomes moved from the cell periphery to the perinuclear region (Fig. 1b, Golgi). This lysosome relocation was also reverted as this cargo left the Golgi complex (Fig. 1b, post-Golgi). Quantitative electron microscopy confirmed such lysosome relocation (Fig. 1c) and lysosome/Golgi ratio increases when the cargo moves from ER to Golgi (ER = 1.91 ± 0.61; Golgi = 3.58 ± 0.52***). Synchronized transport of additional transmembrane or soluble proteins, such as Furin-GFP or Albumin-GFP, performed by DNA microinjection (Supplementary Figure 1a), and galactosyltransferase (GalT), epidermal growth factor receptor (EGFR), or LAMP1, by using the RUSH system[29] (Supplementary Figure 1b), showed a similar lysosome dynamic behaviour. These data thus uncovered a process of lysosome repositioning to the Golgi area that seemed to be regulated by ER-to-Golgi transport.

**Activation of KDELR induces lysosome repositioning**. KDELR has been described as a GPCR[21] that regulates Golgi homeostasis by balancing membrane fluxes between the ER and the Golgi complex[19,20]. Therefore, we hypothesized that the lysosome repositioning upon a load of biosynthetic protein transport in the early secretory route was regulated by KDELR. To test this possibility, we activated the KDELR with a specific ligand, the membrane-permeable peptide KDEL-BODIPY[19,20], which induced a remarkable lysosome relocation to the perinuclear area, as shown by electron microscopy (Fig. 2a) and fluorescence microscopy (Fig. 2b) analyses. Like other GPCRs, KDELR can be activated by its overexpression[19]. Therefore, we separately expressed each of the three KDELR isoforms and found that overexpression of only KDELR1, and not of KDELR2 or KDELR3, produced the relocation of lysosomes to the perinuclear area (Fig. 2c, upper panel). Furthermore, KDELR1-silenced cells did not respond to KDEL-BODIPY (Fig. 2c, bottom panel). Similarly, the expression of a dominant-negative KDELR1 (KDELR-D193N)[19,22] also abrogated lysosome repositioning even in the presence of KDEL-BODIPY (Fig. 2c, upper panel; KDELR2 and KDELR3 dominant-negative forms are not available). These data demonstrate that KDELR1 activation mediates lysosome repositioning regardless of the lack of an overload of cargo secretion. A signalling platform located at the Golgi complex triggered the relocation of peripheral lysosomes, thus revealing a previously undocumented inter-organelle crosstalk. This prompted us to investigate the molecular players involved in the

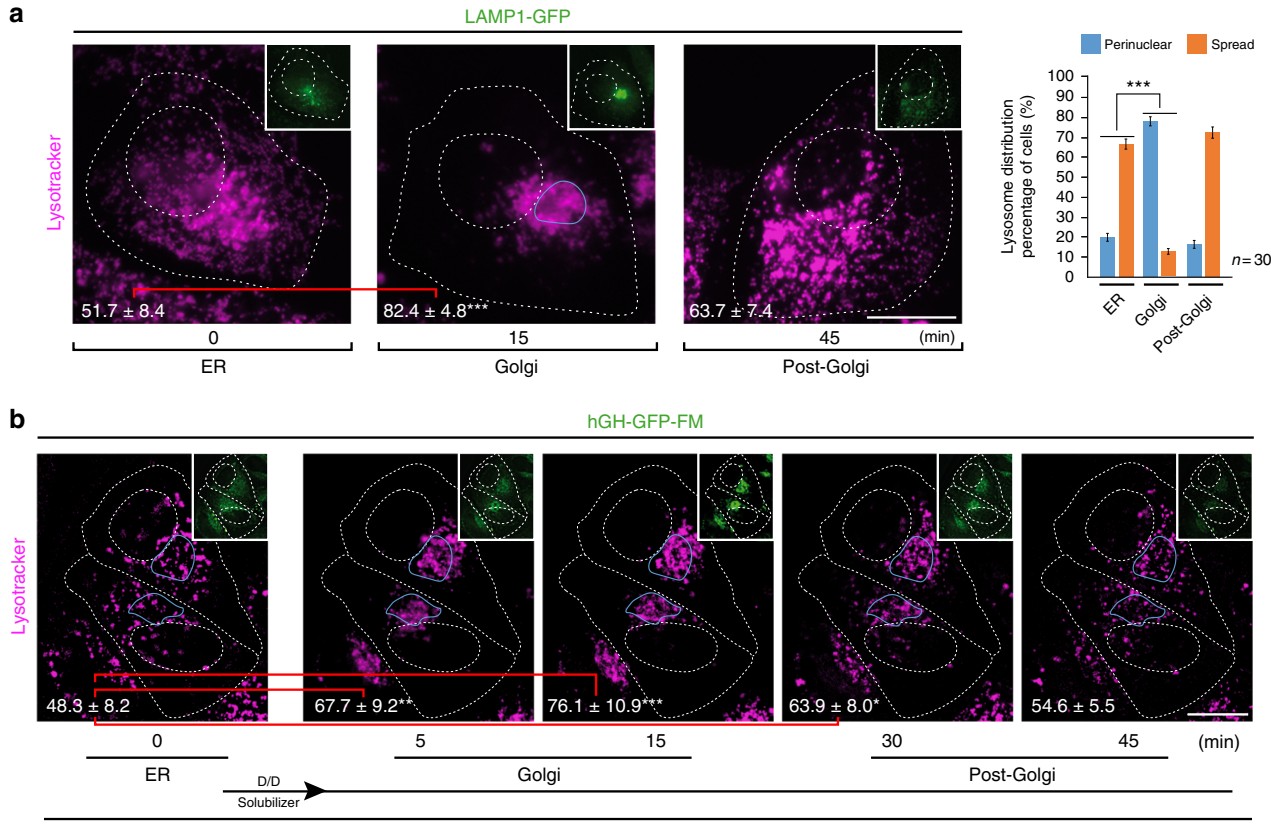

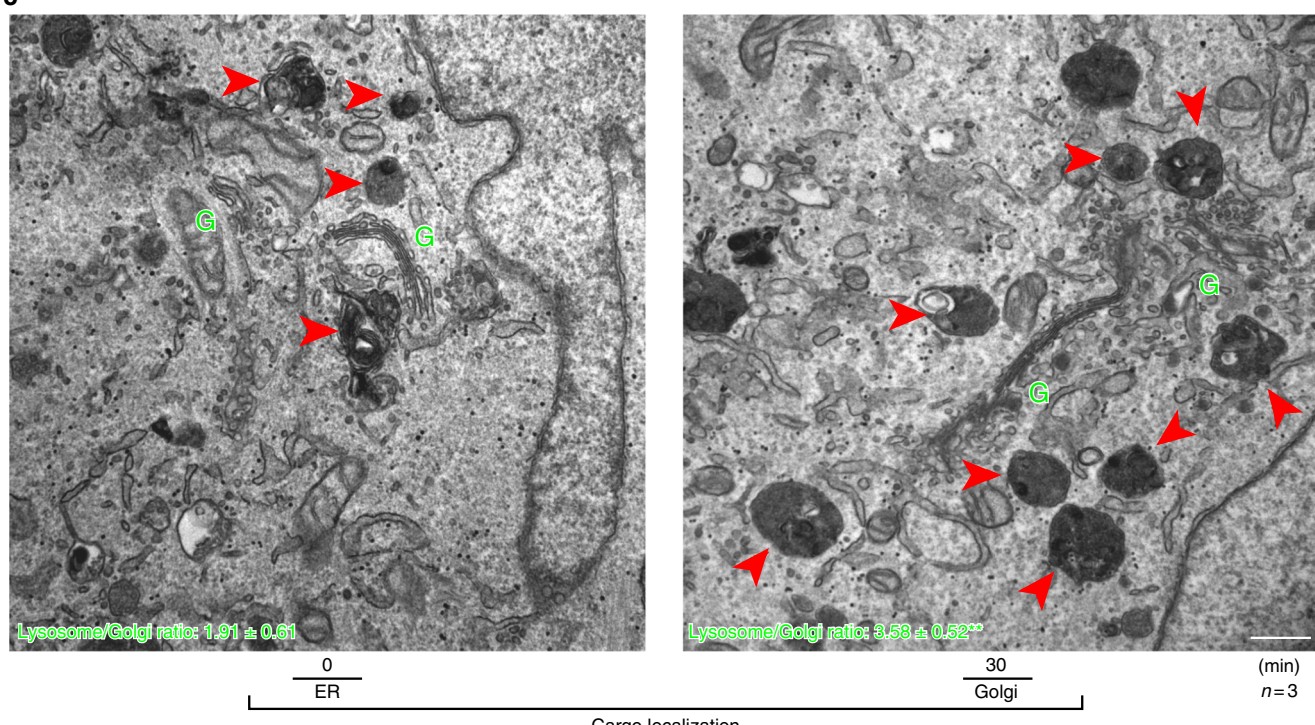

downstream signalling of KDELR, and the functional relevance of such lysosome relocation.

**The Gs/PKA pathway modulates lysosome repositioning**. As a GPCR, KDELR activates the Gs/PKA[19] and Gq/Src pathways[20] at the Golgi complex. To block KDELR downstream signalling, we used membrane-permeable peptides or plasmid-encoded G minigenes, both of which provide short G protein sequences that specifically disrupt either Gs or Gq interactions with GPCRs[19]. Disruption of the KDELR/Gs interaction by the membrane-

**Fig. 1** Endoplasmic reticulum (ER)-to-Golgi cargo transport induces transient lysosome repositioning. **a** HeLa cells were microinjected in the nucleus to deliver a DNA construct encoding LAMP1-GFP (green fluorescent protein) for its synchronized expression and transport from the ER. Representative images show LAMP1-GFP localization in the ER, Golgi or post-Golgi at the indicated time after microinjection. Cells were stained with DeepRed-Lysotracker, and its radial integrated fluorescence intensity was used to quantify the distribution of lysosomes as described in Methods. The graph shows the percentage of cells depicting cytoplasmically spread or perinuclear lysosome distribution ($n = 30$ cells). Scale bar, 10 μm. **b** HeLa cells expressing the human growth hormone fused in tandem to GFP and the polymerization/depolymerization FM domain (hGH-GFP-FM) were subjected to the ER-to-Golgi cargo transport assay. Lysosomes were also stained with DeepRed-Lysotracker. Images were acquired by dual time-lapse to track cargo transport (inset) and lysosome localization before and at the indicated time after addition of D/D solubilizer. The perinuclear distribution of lysosomes was quantified as in **a** and the values are indicated at the bottom of each image ($n = 10$ cells). Scale bar, 10 μm. **c** HeLa cells were examined by electron microscopy when hGH-GFP-FM was localized either at the ER or the Golgi complex. The ratio of lysosomes (red arrowheads) with respect to Golgi stacks (green Gs) was calculated as described in Methods ($n = 3$ independent experiments), and the values are indicated at the bottom of each image (ER = 1.91 ± 0.61; Golgi = 3.58 ± 0.52***). Scale bar, 500 nm. Data are mean ± SEM. *$p < 0.05$; **$p < 0.01$; ***$p < 0.001$ (Student's $t$ tests). All $t$ tests were conducted comparing to control cells

permeable peptide R8-Gs, or by the corresponding Gs minigene, blocked lysosome repositioning induced by KDELR activation, whereas disruption of the KDELR/Gq interaction had no apparent effect (Fig. 3a). Downstream KDELR-dependent signalling relies on the production of cAMP locally at the Golgi complex by adenylyl cyclase 9 (ADCY9)[19]. In agreement with the results shown above, silencing of ADCY9 inhibited KDELR-dependent lysosome repositioning (Fig. 3b). ADCY9-silenced cells incubated with the non-hydrolysable, membrane-permeable cAMP analogue 8-Br-cAMP recovered KDELR-dependent lysosome repositioning, which instead decreased upon incubation with the PKA inhibitors PKI peptide or H89 (Fig. 3c). These data identified the Gs/PKA pathway as the target of the KDELR1 activation in response to cargo loading at the level of the Golgi complex leading to lysosome repositioning.

**DynLRB1 drive lysosome repositioning and secretion from Golgi.** Organelle movement from the periphery to the centre of the cell (i.e. retrograde movement) is mainly driven by dynein-mediated transport along microtubules[30]. As it might be expected, microtubule disruption with nocodazole effectively inhibited the relocation of lysosomes in KDELR-activated cells (Supplementary Figure 2a). We then used short hairpin RNAs (shRNAs) to silence several intermediate/light dynein chains (Supplementary Figure 2b), which are responsible for cargo specificity[30,31]. DynLRB1 was selected because it showed similar lysosome distribution compared to non-silenced cells, but did not relocate lysosomes in response to KDELR activation with KDEL-BODIPY (Fig. 4a), indicating that DynLRB1 is involved in KDELR-mediated lysosome repositioning. Because lysosome repositioning depended on the KDELR-triggered Gs/PKA pathway (Figs. 2, 3), we analysed the presence of putative PKA phosphorylation sites in DynLRB1 that could regulate its function in this process. We found Serine-13 (S13) and S73 as good candidates, but S13 showed a low phosphorylation score (0.5) and was in fact discarded recently as a PKA target[32]. Instead, a large phospho-proteomic analysis identified S73[33], which seems to be effectively phosphorylated by PKA[32]. Therefore, we generated three site-specific DynLRB1 variants to evaluate their effects on lysosome relocation: S73A (phospho-inert), S73D and S73E (phospho-mimetics). Strikingly, expression of either DynLRB1-S73D or DynLRB1-S73E led to lysosome relocation in the absence of KDELR1 activation, while expression of DynLRB1-S73A abolished such relocation even in KDELR1-activated cells (Fig. 4b). Immunogold electron microscopy analysis confirmed the higher number of endo-lysosomal organelles near Golgi stacks in DynLRB1-S73D-expressing cells, as well as the lower number of lysosomes in DynLRB1-S73A-expressing cells (Fig. 4c). Furthermore, expression of DynLRB1-S73A abrogated lysosome relocation induced by KDELR1 overexpression (Fig. 4d), acting most

likely as a dominant-negative version of DynLRB1. In addition, expression of DynLRB1-S73D recovered the lysosome relocation blocked by KDELR1-D193N overexpression (Fig. 4d). Altogether, these data suggest that phosphorylation of DynLRB1 on S73 is necessary for lysosome repositioning in a mechanism that involves KDELR1 activation of Gs/PKA during a load of ER-to-Golgi cargo transport. To evaluate whether lysosome perinuclear repositioning has any role in cargo trafficking, we compared by time-lapse imaging the biosynthetic transport of hGH-GFP-FM in HeLa cells expressing either DynLRB1 wild-type (wt) or DynLRB1-S73A. We found that expression of DynLRB1-S73A delayed cargo exit from the Golgi complex without affecting ER-to-Golgi cargo transport (Fig. 4e). Altogether, these data indicate that lysosome repositioning is linked to the secretion function of the Golgi complex.

**KDELR signalling regulates autophagic flux through DynLRB1.** Lysosome repositioning to the perinuclear area is a hallmark of autophagy activation in response to cellular stressors, including nutritional stress[10]. KDELR1 signalling might then be implicated in autophagy, as it promotes the clearance of harmful and damaging proteins, such as SOD1, A53T α-synuclein and huntingtin, which are involved in neurodegeneration[34]. We thus evaluated the effect of KDELR1 signalling in autophagic flux using the dual-labelled, pH-sensitive autophagy marker LC3 (LC3-GFP-mcherry). By monitoring pH-sensitive GFP and pH-insensitive mcherry emissions, we found that in non-starved cells KDELR1 activation with KDEL-BODIPY increased autophagic flux, effect that was nevertheless reverted by disruption of Gs/PKA with R8-Gs (Fig. 5a). On the other hand, in starved conditions, KDELR1 activation reduced the proportion of cells with autophagosomes (Fig. 5b). This result correlated with a higher autolysosomes/autophagosomes ratio, indicating increased fusion of autophagosomes and lysosomes (Fig. 5c), thus also indicating increased autophagic flux. As it might be expected, Gs/PKA signalling disruption lessened both effects (Fig. 5b, c). These data indicate that KDELR1 signalling activates autophagic flux both in non-starved and starved cells. Next, we assessed the relationship between lysosome redistribution and autophagy. We found that after 30 min of starvation, cells expressing either DynLRB1-S73A or DynLRB1-S73D showed similar numbers of puncta containing the autophagosome marker LC3-GFP (Supplementary Figure 3). However, after 60 min of starvation, cells expressing DynLRB1-S73A showed reduced LC3-GFP signal loss while keeping a similar number of LC3-GFP puncta (Supplementary Figure 3), indicating slower LC3-GFP degradation. Moreover, expression of DynLRB-S73A, which abrogates lysosome redistribution, abolished the increased number of autolysosomes induced by KDELR activation (Fig. 5d–f), and thus the autophagy flux. The opposite condition of expressing DynLRB1-S73D or DynLRB1-S73E,

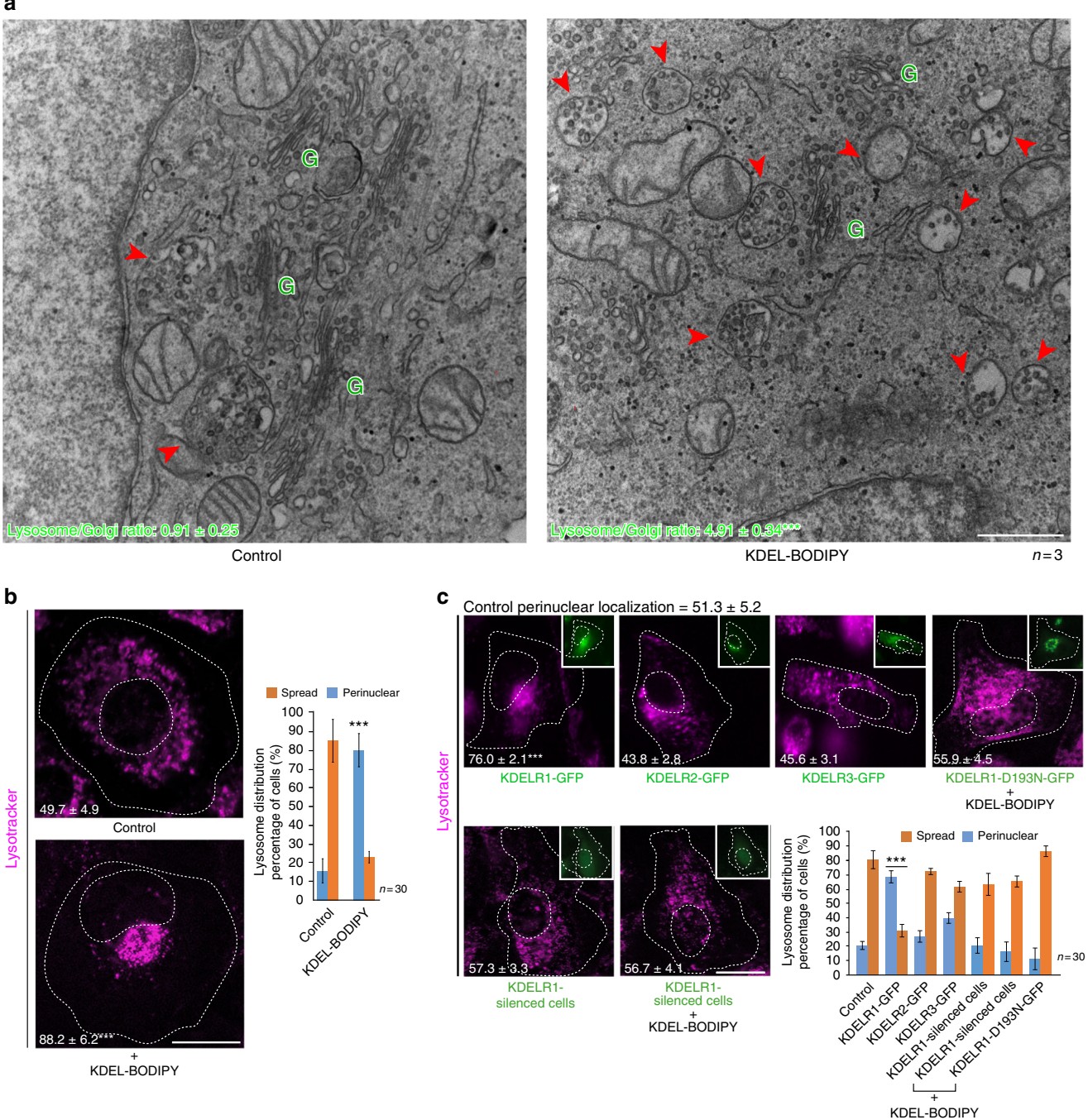

**Fig. 2** Lysosome repositioning is dependent on the KDELR1 isoform. **a** To activate KDELR, HeLa cells were incubated for 30 min with 1 μM KDEL-BODIPY peptide. Cells were fixed with glutaraldehyde and processed for electron microscopy analysis. The ratio of lysosomes (red arrowheads) with respect to Golgi stacks (green Gs) was calculated as described in Methods, and the values are indicated at the bottom of each image (three independent experiments). Scale bar, 500 nm. **b** HeLa cells treated as in **a** were stained with DeepRed-Lysotracker, and its radial integrated fluorescence intensity was used to quantify the distribution of lysosomes as described in Methods. The graph shows the percentage of cells depicting cytoplasmically spread or perinuclear lysosome distribution. The perinuclear distribution of lysosomes was quantified, and the values are indicated at the bottom of each image ($n =$ 30 cells). Scale bar, 10 μm. **c** HeLa cells were transfected (insets) to overexpress either of the indicated green fluorescent protein (GFP)-tagged KDELR isoform or GFP-tagged signalling-defective KDELR1-D193N in the presence of KDEL-BODIPY, or with a short hairpin RNA (shRNA) to silence KDELR1 expression in the absence or presence of KDEL-BODIPY. Lysosomes were stained and their distribution quantified as indicated in **b**. The graph shows the percentage of cells depicting cytoplasmically spread or perinuclear lysosome distribution. The perinuclear distribution of lysosomes was quantified, and the values are indicated at the bottom of each image ($n =$ 30 cells). Scale bar, 10 μm. Data are means ± SEM. ***$p < 0.001$ (Student's $t$ tests). All $t$ tests were conducted comparing to control cells

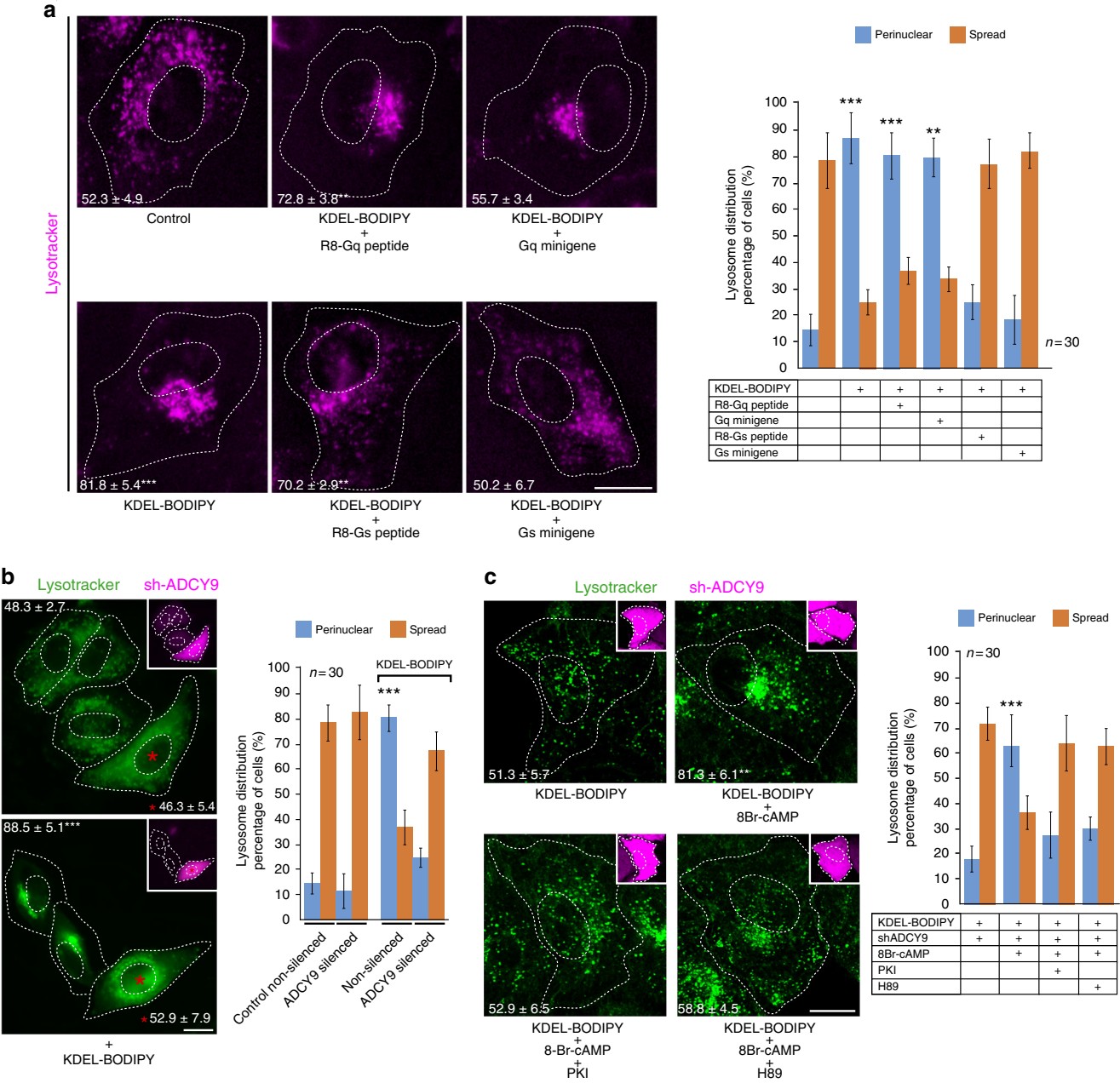

**Fig. 3** The KDELR-dependent cAMP/PKA signalling pathway regulates lysosome repositioning. **a** HeLa cells were left untreated (Control) or incubated for 15 min at 37 °C with 1 μM KDEL-BODIPY peptide, to activate KDELR, in the absence (KDEL-BODIPY) or presence of either 10 μM R8-Gq peptide (KDEL-BODIPY + R8-Gq peptide) or 10 μM R8-Gs peptide (R8-Gs peptide), or in conjunction to transfection for the expression of a Gq minigene (KDEL-BODIPY + Gq minigene) or a Gs minigene (KDEL-BODIPY + Gs minigene). Cells were stained with DeepRed-Lysotracker, and its radial integrated fluorescence intensity was used to quantify the distribution of lysosomes as described in Methods. The graph shows the percentage of cells depicting cytoplasmically spread or perinuclear lysosome distribution. The perinuclear distribution of lysosomes was quantified, and the values are indicated at the bottom of each image ($n = 30$ cells). **b** HeLa cells stably transfected with sh-ADCY9 (insets; red fluorescent protein (RFP)-positive cells; red asterisks) were incubated with Lysotracker green and then either left untreated (Control) or treated with KDEL-BODIPY peptide as in **a**. Lysosomes distribution was quantified as indicated in **a**. The graph shows the percentage of cells depicting cytoplasmically spread or perinuclear lysosome distribution. The perinuclear distribution of lysosomes was quantified, and the values are indicated at the bottom of each image ($n = 30$ cells). **c** ADCY9-silenced HeLa cells (insets; RFP-positive cells) were incubated with KDEL-BODIPY alone as in **a** or co-incubated with only the non-hydrolysable cAMP analogue 8-Br-cAMP (100 μM) or with 8-Br-cAMP in conjunction to either of the PKA inhibitors PKI (50 μM) or H89 (30 μM). Lysosome distribution was quantified as indicated in **a**. The graph shows the percentage of cells depicting cytoplasmically spread or perinuclear lysosome distribution. The perinuclear distribution of lysosomes was quantified, and the values are indicated at the bottom of each image ($n = 30$ cells). Scale bar, 10 μm. Data are means ± SEM. ***$p < 0.001$ (Student's $t$ tests). All $t$ tests were conducted comparing to control cells

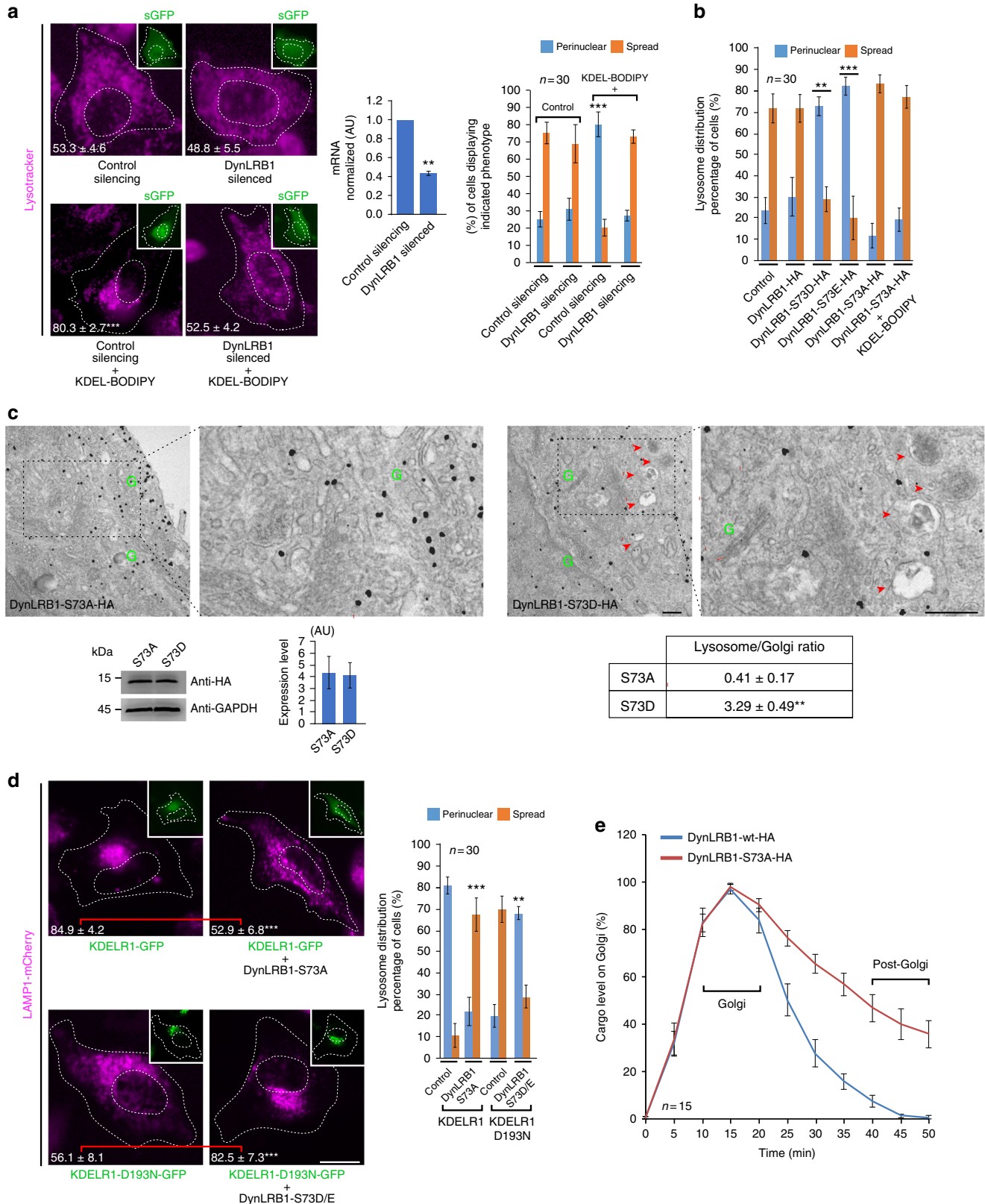

which promotes lysosome perinuclear redistribution in the absence of KDELR activation, resulted in a robust increase in autophagic flux (Fig. 5g–i). Therefore, autophagy progression during KDELR activation requires lysosome redistribution at sites of autophagosome formation, which is a possibility that needs further investigation.

**Exocytic trafficking and autophagy are mutually regulated**. Our results suggested a relationship between autophagy and biosynthetic secretion. To analyse the role of autophagy in cargo exit from the Golgi complex, we inhibited autophagosome formation either pharmacologically with SAR405[35] or genetically by silencing ATG5[36,37]. Both SAR405 (Supplementary Figure 4) or

**Fig. 4** LRB1 dynein light chain drives the lysosome repositioning that is necessary to sustain secretion. **a** HeLa cells subjected either to control silencing or DynLRB1 silencing (insets; green fluorescent protein (GFP)) were left untreated or incubated for 15 min with 1 μM KDEL-BODIPY peptide to induce lysosome repositioning. Silencing was controlled by determining the messenger RNA (mRNA) level of DynLRB1 (first graph, $n = 3$ independent experiments). Alternatively, cells were stained with DeepRed-Lysotracker, and its radial integrated fluorescence intensity was used to quantify the distribution of lysosomes as described in Methods. The percentage of cells depicting cytoplasmically spread or perinuclear lysosome distribution was calculated ($n = 30$ cells). The perinuclear distribution of lysosomes was quantified, and the values are indicated at the bottom of each image ($n = 30$ cells). Scale bar, 10 μm. **b** HeLa cells not transfected (Control) or expressing either of the indicated HA-tagged variants of DynLRB1 were either left untreated or incubated with KDEL-BODIPY. Lysosomes were stained, and their distribution quantified as indicated in **a**, ($n = 30$ cells). **c** HeLa cells were transfected to express either of the indicated hemagglutinin (HA)-tagged variants of DynLRB1 and processed for immunogold labelling (10 nm) against the HA-tag and subsequent electron microscopy analysis. DynLRB1 expression was assessed by western blotting using anti-HA or anti-glyceraldehyde 3-phosphate dehydrogenase (GAPDH) antibodies, and the corresponding quantification is shown in a bar graph (data are means ± SEM). The ratio of lysosomes (red arrowheads) with respect to Golgi stacks (green Gs) was calculated as described in Methods, and the values are indicated in the table depicted in the fourth image ($n = 3$ independent experiments). The second and fourth images correspond to a two-fold magnification of the indicated regions of the first and third images, respectively. **d** Cells expressing LAMP1-mcherry and either KDELR1-GFP or KDELR-D193N-GFP were left without further treatment (Control) or transfected to co-express either of the indicated HA-tagged variants of DynLRB1. Lysosomes were stained, and their distribution quantified as indicated in **a**, ($n = 30$ cells). Scale bar, 10 μm. **e** HeLa cells expressing human growth hormone fused to the polymerization/depolymerization FM domain (hGH-GFP-FM) and co-expressing either DynLRB1-wt-HA (Control) or DynLRB1-S73A-HA were subjected to the ER-to-Golgi transport assay, and images were acquired for 50 min at 5 min interval. The level of cargo on the Golgi complex was quantified ($n = 15$ cells). Data are means ± SEM. $**p < 0.01$; $***p < 0.001$ (Student's $t$ tests). All $t$ tests were conducted comparing to control cells. wt Wild type

ATG5 silencing strongly reduced cargo exit from the Golgi complex, as shown by fluorescence time-lapse imaging (Fig. 6a), and biochemical analysis (Fig. 6b). These data indicate that sustained transport of a load of cargo from the Golgi complex to the PM requires autophagy. To have additional evidence of the function of autophagy in this process, we tested the role of p62/SQSTM1, which is a selective autophagy cargo receptor[38]. A previous genome-wide RNA interference screening revealed that p62/SQSTM1 silencing reduces the secretion of the traffic reporter VSVG[39]. Thus, we evaluated the role of p62/SQSTM1 in the repositioning of lysosomes during secretion. We first analysed the distribution of p62/SQSTM1 during ER-to-Golgi-to-PM cargo transport in non-starved cells. We found that the synchronized release from the ER of the exocytic transport reporter hGH-GFP-FM correlated with increased number of p62/SQSTM1-positive puncta at the time of cargo arrival at the Golgi complex (Fig. 7a). Importantly, activation of KDELR by KDEL-BODIPY triggered a similar response of p62/SQSTM1, which was inhibited by KDELR1 silencing (Supplementary Figure 5). These results indicate that activation of KDELR signalling generates a response of p62/SQSTM1 that could be related to its function in autophagy.

To further analyse the role of p62/SQSTM1, we decided to generate p62/SQSTM1 mutation variants and test whether their expression would reveal some regulatory function in the repositioning of lysosomes during secretion. We took advantage of the fact that p62/SQSTM1 interacts with LC3 and with the dynein motor complex, both interactions through a region encompassing residues 177–194, which when overexpressed as a short p62/SQSTM1 peptide (p62/SQSTM1$^{177–194}$) blocks the degradation function of p62/SQSTM1[40]. Moreover, a large-scale phospho-proteomic analysis showed that p62/SQSTM1 could be phosphorylated on S182[33]. Therefore, we generated and evaluated the effects of the phospho-mutant versions S182A and S182E. The comparison of these phospho-mutants and p62/SQSTM1-wt showed that their responses under various conditions were very similar: they were recruited to LC3-GFP puncta after 30 min of starvation (Supplementary Figure 6a), they exchanged dynamically between autophagic puncta and cytosolic pools (Supplementary Figure 7), they interacted with LC3-GFP (Supplementary Figure 6c) and they formed oligomers (Supplementary Figure 6d), which are important for the interaction with LC3[41]. These results indicate that these phospho-mutants can be used to evaluate the role of S182 in the function of p62/SQSTM1. In fact, although the

responses of the phospho-mutants were similar, we found differences that could be ascribed to the phospho-inert or phospho-mimetic character of the corresponding amino acid substitution. This is illustrated by reduced decrease of LC3-GFP puncta (Supplementary Figure 6a) and slower LC3-GFP degradation (Supplementary Figure 6b) in cells expressing p62/SQSTM1-S182A, suggesting that phosphorylation of S182 on p62/SQSTM1 is necessary for autophagic flux, and that p62/SQSTM1-S182A is acting as a dominant-negative mutant. We thus concluded that these phospho-mutants could be used to assess the function of p62/SQSTM1 in the protein secretion-induced repositioning of lysosomes.

Similar to the response of endogenous p62/SQSTM1 (Fig. 7a), a load of hGH-GFP-FM transport from the ER to the Golgi complex resulted in higher number and increased fluorescence intensity of puncta bearing either overexpressed p62/SQSTM1-wt (Fig. 7b, upper panel) or overexpressed p62/SQSTM1-S182A (Fig. 7b, bottom panel). Concomitant to cargo exit from the Golgi complex to a post-Golgi site, we detected that the number and fluorescence intensity of p62/SQSTM1-wt puncta were reduced (Fig. 7b, upper panel), as expected for p62/SQSTM1-wt degradation if those were autophagic puncta (i.e. autophagosomes) that fused with lysosomes. In contrast, the number and fluorescence intensity of p62/SQSTM1-S182A puncta remained unaffected (Fig. 7b, bottom panel), suggesting a reduced degradation of this mutant protein. Importantly, cells expressing p62/SQSTM1-S182A also showed impaired exit of hGH-GFP-FM from the Golgi complex (Fig. 7b). This outcome seemed to be a broad effect on the secretory route at the Golgi level because the transport from the Golgi complex to the PM of other cargo such as VSVG and LDLR was also reduced (Supplementary Figure 8c-d). In addition, we found that at the time of cargo arrival at the Golgi complex, p62/SQSTM1-wt puncta co-localized with lysosomes to a higher extent compared to p62/SQSTM1-S182A puncta, and that the opposite effect occurred at the time of cargo localization in post-Golgi compartments (Fig. 7c). In contrast, at steady-state level, p62/SQSTM1-S182A or p62/SQSTM1-S182E puncta showed co-localization with lysosomes to a lower or higher extent, respectively, compared to p62/SQSTM1-wt puncta (Fig. 7d). These data further indicate that p62/SQSTM1 is necessary for efficient Golgi-to-PM cargo transport, and also further suggest that a load of secretory biosynthetic cargo activates autophagy.

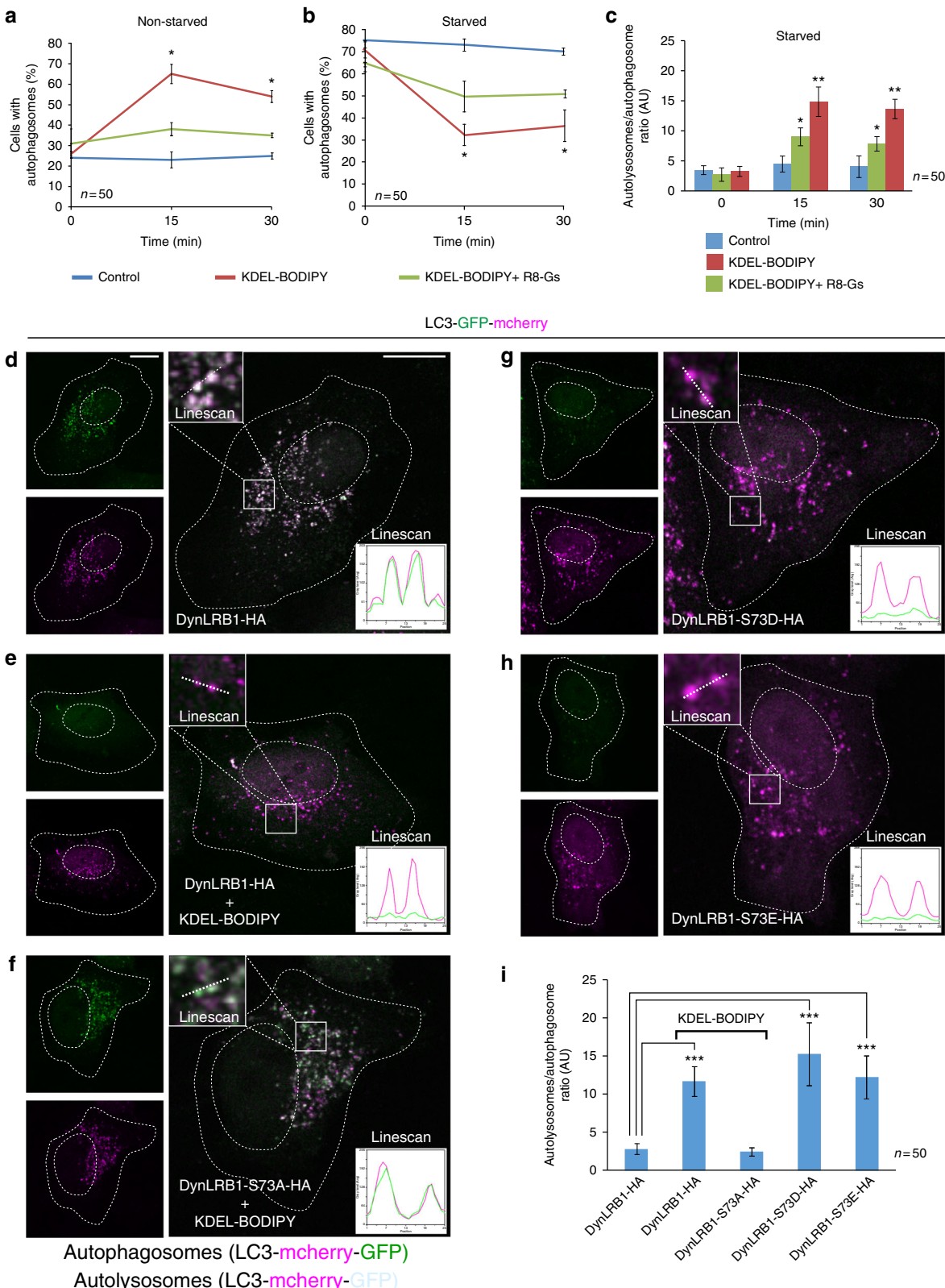

Autophagosomes (LC3-mcherry-GFP)
Autolysosomes (LC3-mcherry-GFP)

The coordinated response of lysosomes, autophagosomes and the Golgi complex revealed here suggested the possibility of its impact on secretion-mediated physiological processes. To test this hypothesis, we explored its relevance in the secretion of endogenous interleukin-2 (IL-2) during T cell activation. We found that either the blocking of KDELR signalling (Fig. 7e) or

the expression of p62/SQSTM1-S182A (Fig. 7f) greatly impaired IL-2 secretion in T-lymphocytes activated by antigen presentation. These results strongly suggest that the KDELR signalling that triggers lysosome repositioning and activation of autophagy regulates relevant physiological secretion processes such as those that operate in the immune system.

**Fig. 5** The fusion of autophagosomes and lysosomes is regulated by KDELR and DynLRB1. **a** H4 cells transfected to express the pH-sensitive autophagy flux marker LC3 (mcherry-GFP-LC3) were left untreated (Control) or incubated for 15 or 30 min at 37 °C with 1 μM KDEL-BODIPY peptide alone, or in combination with 10 μM R8-Gs peptide. The graph depicts the quantification of cells with autophagosomes that were regarded as puncta showing mcherry and green fluorescent protein (GFP) fluorescence ($n = 50$ cells). **b** H4 cells transfected as in **a** were subjected to nutrient starvation for 1 h to induce autophagosome formation, and then incubated with cell-permeable peptides as in **a**. The graph depicts quantification of cells ($n = 50$ cells) with autophagosomes as indicated in **a**. **c** Quantification of the ratio between autolysosomes and autophagosomes in cells treated as indicated in **b**. Autolysosome/autophagosome ratio was calculated by automatic counting of red (mcherry, pH-insensitive) and green (GFP, pH-sensitive) positive puncta ($n = 50$ cells). **d**–**h** HeLa cells were co-transfected to express mcherry-GFP-LC3 and either of the indicated DynLRB1-HA (hemagglutinin) variants, and either left untreated (**d**, **g** and **h**) or incubated for 30 min with 1 μM KDEL-BODIPY (**e** and **f**). Insets show linescans (25-length × 4-width pixels) of representative regions depicting the presence of red and green fluorescent emission profiles of autophagic puncta. Red and green emission identifies puncta as autophagosomes, and lack of green emission identifies puncta as autolysosomes ($n = 3$ independent experiments). Scale bars, 10 μm. **i** Quantification of the ratio between autolysosomes and autophagosomes as indicated in **c** in cells ($n = 50$ cells) treated as indicated in **d**–**h**. Data are means ± SEM. *$p < 0.05$; ***$p < 0.001$ (Student's $t$ tests). All $t$ tests were conducted comparing to control cells

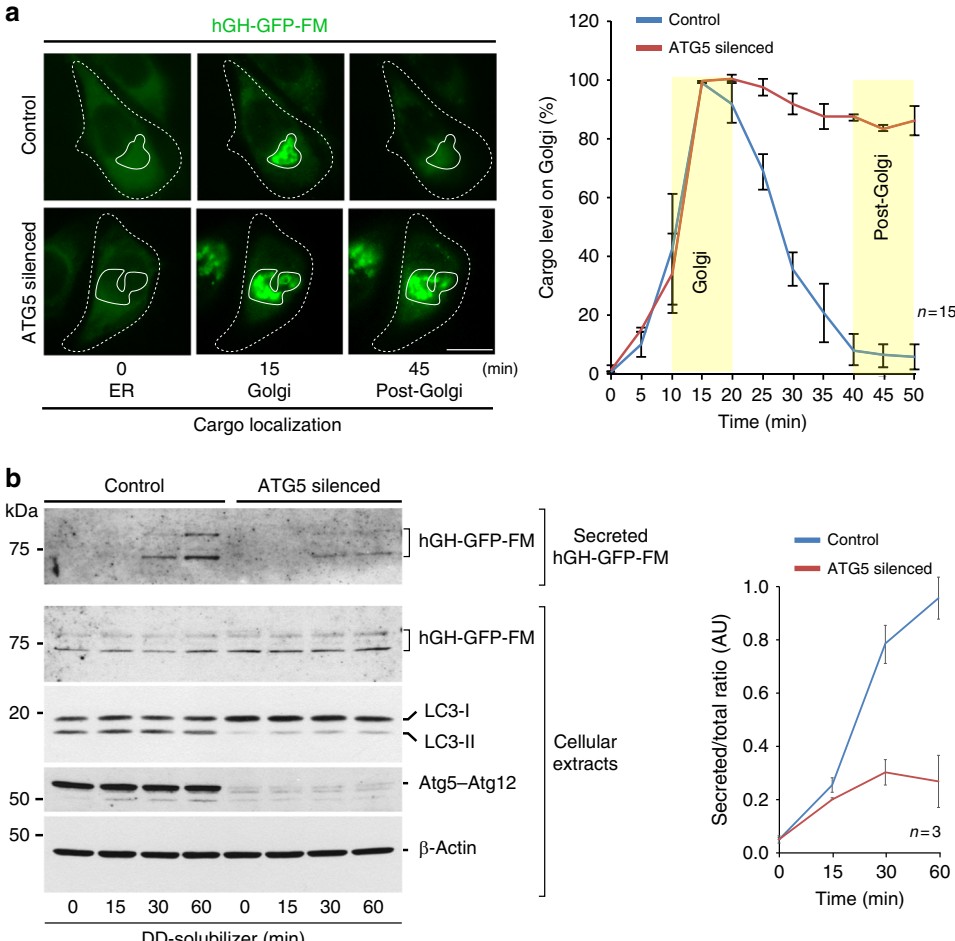

**Fig. 6** ATG5-dependent autophagy is required to sustain protein secretion. **a** HeLa cells expressing human growth hormone fused to the polymerization/depolymerization FM domain (hGH-GFP-FM) were subjected to control silencing (Control) or ATG5 silencing and further subjected to the endoplasmic reticulum (ER)-to-Golgi cargo transport assay. Cargo trafficking was tracked by time-lapse fluorescence microscopy. Representative images show hGH-GFP-FM localization in the ER, Golgi or post-Golgi at the indicated times after addition of DD-solubilizer ($n = 15$ cells). Scale bar, 10 μm. The graph depicts the quantification of the level of cargo at the Golgi complex during the time of cargo trafficking from images as those shown in **a**, highlighting in yellow the timeframe of typical cargo localization in the Golgi complex or post-Golgi **b**. HeLa cells expressing hGH-GFP-FM were subjected to the same treatments shown in **a**, and the medium (Secreted) and cellular extracts were collected at the indicated times after addition of DD-solubilizer. Proteins from the medium and cellular extracts were analysed by sodium dodecyl sulphate-polyacrylamide gel electrophoresis (SDS-PAGE) and western blotting using antibodies to the proteins indicated on the right. The graph depicts the quantification of the ratio between secreted and total hGH-GFP-FM detected by western blotting ($n = 3$ independent experiments). Data are means ± SEM. GFP green fluorescent protein

**Lipid-droplet turnover is needed to sustain protein secretion.** We reasoned that if lysosome repositioning during secretory biosynthetic overload engaged ATG5-mediated autophagy, other related adaptive cellular processes might also be involved.

Interestingly, autophagy plays key roles during lipid droplet (LD) turnover, and vice versa[42]. In this regard, a curious observation is that p62/SQSTM1 interacts with LDs[43]. Importantly, LD turnover is also integrated with the secretory route, including

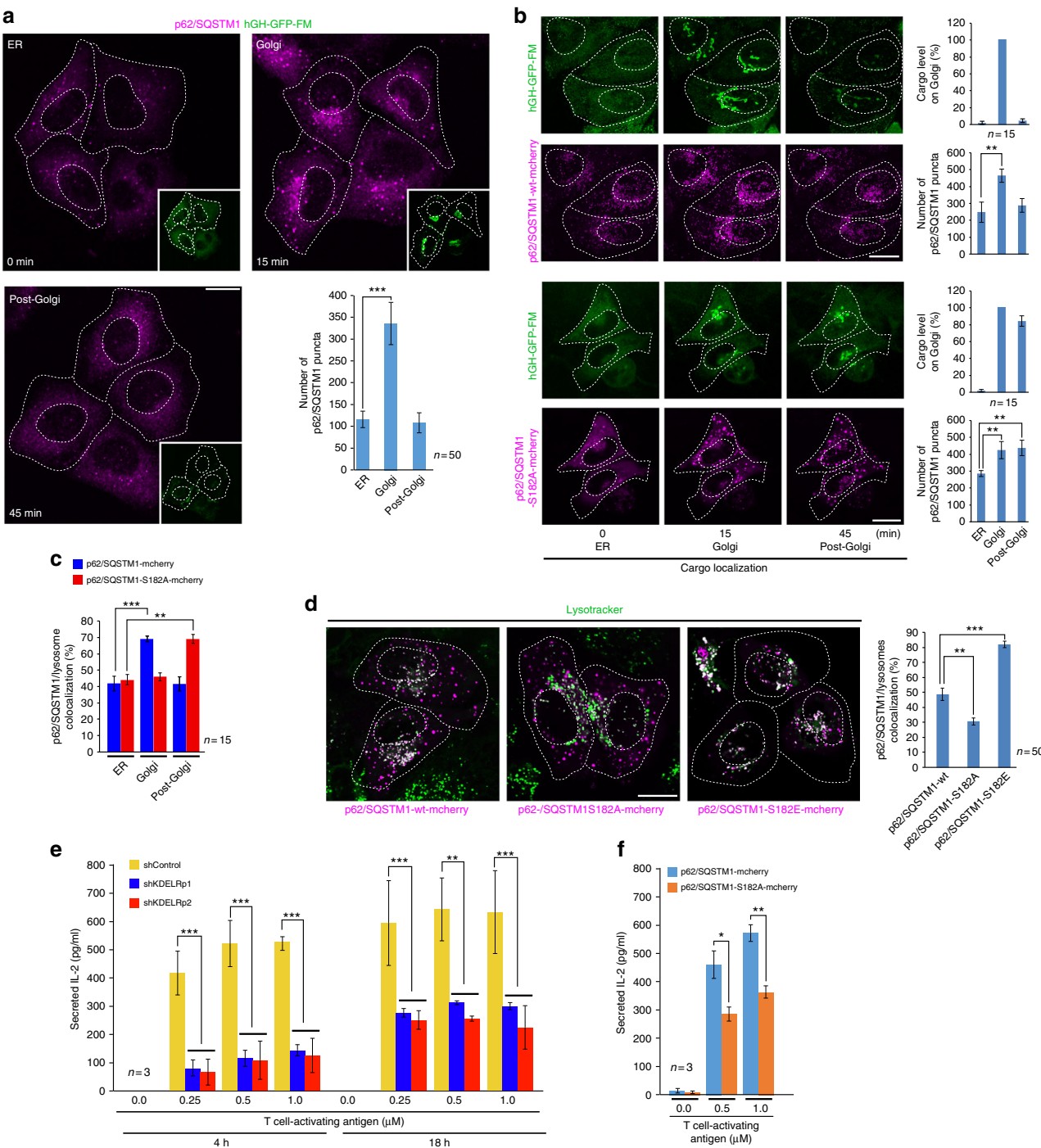

processes at the Golgi complex[44]. Thus, these data led us to investigate the relationship between KDELR signalling and autophagy-mediated LD turnover.

We found that overexpression of either KDELR1, p62/SQSTM1-wt[45] or phospho-mimetic p62/SQSTM1-S182E, shown above to enhance autophagic flux, resulted also in reduced number of LDs (Fig. 8a, b). By contrast, overexpression of KDELR1-D193N or p62/SQSTM1-S182A, as well as KDELR1 silencing, increased the number of LDs (Fig. 8a, b). We also detected LDs closely apposed to p62/SQSTM1 puncta (Fig. 8c) and found that in many cases they moved together as if they were interacting (Movie S1). We analysed quantitatively the contacts between p62/SQSTM1 and LDs, as well as the speed of

their joint movement. Compared to p62/SQSTM1-wt puncta, p62/SQSTM1-S182A puncta showed a fewer number of contacts to LDs (Fig. 8d, bottom panel; Movie S3), whereas p62/SQSTM1-S182E puncta showed more, which correlated with increased speed of joint movement (Fig. 8d, upper panel; Movie S2). Because the interaction of p62/SQSTM1 puncta and LDs was affected by phospho-mimetic substitutions of S182, which is an amino acid contained in a region of p62/SQSTM1 involved in the interaction with dynein[40], we tested the effect of the DynLRB1 phospho-mimetic mutants. We detected fewer contacts between p62/SQSTM1-wt puncta and LDs in cells co-expressing DynLRB1-S73A (Fig. 8e, upper panel; Movie S4). In contrast, co-expression of DynLRB1-S73D increased the number of

**Fig. 7** The autophagy cargo receptor p62/SQSTM1 is required to sustain protein secretion during lysosome repositioning. **a** HeLa cells expressing human growth hormone fused to the polymerization/depolymerization FM domain (hGH-GFP-FM) were subjected to the endoplasmic reticulum (ER)-to-Golgi cargo transport assay and then subjected to immuno-staining to detect endogenous p62/SQSTM1. Representative images show p62/SQSTM1 puncta distribution at the indicated times of hGH-GFP-FM localization in the ER, Golgi or post-Golgi (insets) after addition of DD-solubilizer. The graph depicts the quantification of the number of p62/SQSTM1 puncta at the times indicated in the images ($n = 50$ cells). **b** HeLa cells expressing hGH-GFP-FM were transfected to co-express p62/SQSTM1-wt-mcherry or the p62/SQSTM1-S182A-mcherry mutant. After 16 h, cells were subjected to the ER-to-Golgi cargo transport assay. Representative images show the localization of hGH-GFP-FM and the distribution of puncta containing either of the p62/SQSTM1-mcherry variant, in conditions as those shown in **a**. The graphs depicts either the quantification of hGH-GFP-FM in the Golgi complex or the quantification of the number of puncta containing either of the p62/SQSTM1-mcherry variant at the times indicated in the images ($n = 15$ cells). **c** HeLa cells transfected to express either of the indicated p62/SQSTM1-mcherry variants were incubated with Lysotracker. Cells were nutrient-starved by incubation for 1 h in Hank's balanced salt solution. At the indicated times after, images were acquired to calculate the level of co-localization between lysosomes and puncta containing either of the indicated p62/SQSTM1-mcherry variant ($n = 15$ cells). **d** HeLa cells were transfected and incubated with Lysotracker as in **c**, but maintained in regular culture medium. Images were acquired to calculate the level of co-localization in steady-state conditions between lysosomes and puncta containing either of the indicated p62/SQSTM1-mcherry variant. The quantification is shown in the graph at the right of the panel ($n = 50$ cells). **e** Mouse LMR7.5 T-lymphocytes were subjected to control silencing (shControl) or silencing of KDELR1 using either of the indicated short hairpin RNA (shRNA). Cells were subjected to an antigen-peptide presentation assay using the indicated concentrations of antigen for the indicated periods of time. To determine T-lymphocyte activation, the level of secreted interleukin-2 (IL-2) was estimated using an enzyme-linked immunosorbent assay (ELISA) assay ($n = 3$ independent expriments). **f** Mouse LMR7.5 T-lymphocytes were transfected to express p62/SQSTM1-mcherry-wt or p62/SQSTM1-mcherry-S182A. After transfection, cells were subjected to an antigen-peptide presentation assay and to estimation of T-lymphocyte activation as shown in **e** ($n = 3$ independent experiments). Data are means ± SEM. Scale bar, 10 μm. *$p < 0.05$; **$p < 0.01$; ***$p < 0.001$ (Student's $t$ tests). wt Wild type

contacts and, more notably, the speed of joint movement (Fig. 8e, bottom panel; Movie S5). These data suggest that DynLRB1 plays a role in p62/SQSTM1-mediated LD turnover. We confirmed such a role by silencing DynLRB1 expression, which led to an increased number and content of LDs, whereas DynLRB2 silencing had no effect (Fig. 8f). Furthermore, overexpression of DynLRB1-S73D reduced the number of LDs (Fig. 8f). We conclude that the functions revealed for DynLRB1 and p62/SQSTM1 in the repositioning of lysosomes are cooperating in LD turnover, which at the same time is necessary for exit of cargo overload from the Golgi complex. As S73D is a PKA phosphorylated mimicking motif[32,33], we tested the effects of PKA activation by either the non-hydrolysable cAMP analogue 8-Br-cAMP (Fig. 8g) or KDELR-mediated activation with KDELR ligand (KDEL-BODIPY?)[19] (Fig. 8h). Both conditions decreased the content of LDs, while pharmacological PKA inhibition partially reverted this effect (Fig. 8g, h). The expression of p62/SQSTM1-S182A also reduced the effects of PKA activation on LDs. This was more evident on KDELR-activated cells. Therefore, LDs might have a role in secretion.

We next analysed the relationship between LD turnover and lysosome repositioning during cargo transport in cells impaired of ATG5-dependent autophagy. ATG5-silenced cells exhibited a reduced number of LDs and of their content (Fig. 9a), in agreement with a previous report that proposes ATG5-dependent autophagy as requirement for LD generation[46]. Interestingly, we found that fatty acid supplementation with palmitic acid recovered the number and content of LDs in ATG5-silenced cells to near the level of control cells (Fig. 9a), and strikingly also rescued cargo transport out of the Golgi complex (Fig. 9b). These results are consistent with the notion that decreased autophagy impairs the supply of fatty acids required to maintain a balanced LD turnover necessary to sustain cargo transport in the secretory route. We obtained further evidence of the requirement of LD turnover by overexpressing in ATG5-silenced cells either of the phospho-inert mutants DynLRB1-S73A and p62/SQSTM1-S182A, both shown above that act decreasing autophagy and lysosome repositioning. In both conditions, palmitic acid was not able to efficiently rescue cargo transport out of the Golgi complex (Fig. 9b). These results suggest that conditions that re-establish LD generation are not enough to rescue impaired secretion when lysosome repositioning is precluded. Together, the results shown here indicate that a load of biosynthetic secretory transport

triggers a dynamic cellular response that involves the turnover of LD and the lysosomal degradation pathway.

## Discussion

Herein, we provide evidence that KDELR activation and signalling through the Gs/PKA pathway promoted relocation of lysosomes from the cell periphery to the Golgi region and enhanced both autophagy and LD consumption, altogether converging towards the maintenance of Golgi-to-PM trafficking. Inhibition of lysosomal relocation or of autophagy reduced the exit of protein cargo from the Golgi complex. Similar secretion impairment occurred under autophagy-dependent conditions that perturbed LD turnover. Because lysosomes constitute the compartment where autophagy takes place, and are also the hosts of a nutrient-sensor machinery that regulates proteostasis[25,47,48], our observations reveal a previously unsuspected mechanism linking metabolism to Golgi-mediated secretion. Therefore, lysosomes emerge as novel sensors of a signalling pathway triggered by protein transport at the *cis*-Golgi whereby the KDELR undergoes activation. Lysosomes acquire then a function: regulation of Golgi-dependent secretion.

Previous studies posit KDELR1 as an early GPCR sensor of ER-to-Golgi trafficking activity through a Gs/PKA signalling pathway[19]. The functional meaning of these Golgi-generated PKA signals had remained little understood. We add here that KDELR-triggered Gs/PKA signalling impinges upon what we called the TIDeRS. This response represents an example of inter-organelle crosstalk engaging distinct machineries in a specific task: the adjustment of membrane trafficking activity out of the Golgi to metabolic needs and membrane trafficking into the Golgi. In the adjustment of Golgi input/output membrane trafficking, TIDeRS includes lysosome/autolysosome centripetal relocation accompanied by autophagy, which includes enhanced LD degradation, all being requirements for membrane trafficking from the Golgi complex to the plasma membrane.

To analyse different aspects of TIDeRS, we developed phospho-defective mutants of DynLRB1 and p62/SQSTM1 that perturbed lysosome repositioning, autophagic flux and the exit from the Golgi complex of a secretion cargo (see Supplementary Figure 9 for a model). The p62/SQSTM1 mutant involved the S182 residue within the 177–194 polypeptide region previously identified as a dynein-interacting domain[40]. We found this S182 residue to be critical for Golgi-to-PM cargo transport. The

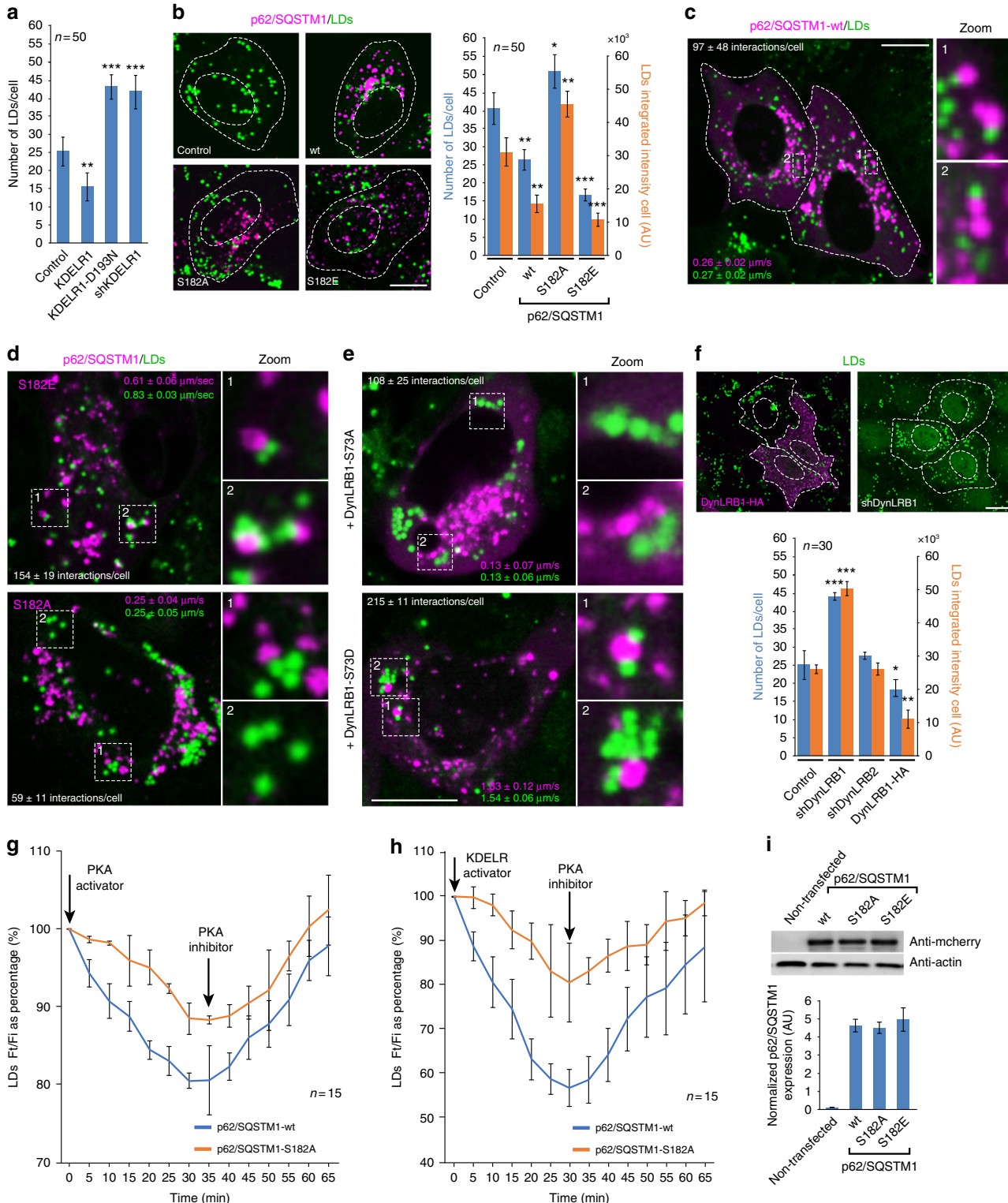

mechanism very likely is related to p62/SQSTM1-promoted fusion of lysosomes and autophagosomes, and the subsequent autophagy-mediated degradation of LDs. The phospho-defective version of either DynLRB1 or p62/SQSTM1 impaired LD degradation through a KDELR1- and PKA-dependent mechanism, resulting in cargo exit blocks at the Golgi complex.

Lipids are central to cellular survival and viability, participating in processes that maintain cell structure, organelle function and energy homeostasis. During nutrient stress, LD turnover contributes with fatty acids that sustain ATP production[46,49]. Our

findings support a role of DynLRB1 and p62/SQSTM1 on LD turnover controlled by PKA and KDELR. This likely impinges upon energy-consuming processes during membrane transport along the secretory route. The exact functional relationship between fatty acid metabolism and an overload of biosynthetic secretory cargo remains to be elucidated.

Cells respond constantly and rapidly to a wide range of internal and external signals that maintain homeostasis and avoid cellular damage[10,17,18]. Such plasticity is essential for the adaptation of specific functions to cellular demands and includes changes in the

**Fig. 8** KDELR1 modulate lipid droplet turnover during activation of lysosome repositioning. **a** HeLa cells untreated (Control) or transfected with KDELR1, KDELR1-D193N or short hairpin RNA (shRNA) against KDELR1 were fixed and lipid droplets (LDs) were stained and a quantitative analysis of the number of LDs was calculated ($n = 50$ cells). **b** H4 cells untreated (Control) or transfected to express either of the indicated p62/SQSTM1-mcherry variants and LDs were stained and quantified ($n = 50$ cells). **c** H4 cells were transfected to transiently express p62/SQSTM1-mcherry-wt (wild type), followed by LD staining. Live-cell images were acquired, and the number of interactions between LDs and p62/SQSTM1-mcherry-wt puncta and of the joint speeds of movement were calculated as indicated in Methods ($n = 3$ independent experiments). **d** H4 cells were transfected to transiently express p62/SQSTM1-mcherry-S182E or p62/SQSTM1-mcherry-S182A, followed by LD staining and live-cell imaging acquisition and analysis ($n = 3$ independent experiments) as indicated in **c**. **e** H4 cells were transfected to transiently co-express p62/SQSTM1-mcherry-wt and either of the indicated hemagglutinin (HA)-tagged DynLRB1 variants, followed by LD staining and live-cell imaging acquisition and analysis as indicated in **c**. The number of interactions and the joint speeds of movement ($n = 3$ independent experiments) was calculated. **f** HeLa cells were transfected either to express DynLRB1-HA-wt (DynLRB1-HA) or with an shRNA against DynLRB1 (shDynLRB1). Cells were fixed and LDs were stained and immunofluorescence to detect DynLRB1-HA-wt. The graph shows the quantification of the number and fluorescence intensity of LDs of cells subjected to the indicated treatments ($n = 30$ cells). **g** H4 cells were transiently transfected to express p62/SQSTM1-wt or p62/SQSTM1-S182A and were stained for LDs, and images were acquired every 5 min under PKA activation and subsequent PKA inhibition ($n = 15$ cells). (**h**) H4 cells were transiently transfected to express p62/SQSTM1-wt or p62/SQSTM1-S182A and were stained for LDs, with images acquired every 5 min under KDELR activation and subsequent PKA inhibition ($n = 15$ cells). **i** H4 cells transfected as in **b–h** were subjected to sodium dodecyl sulphate-polyacrylamide gel electrophoresis (SDS-PAGE) and western blot against actin (load control) and mCherry (p62/SQSTM1). Normalized mCherry/actin signal is shown on the plot bars ($n = 3$ independent experiments). Data are means ± SEM. *$p < 0.05$; **$p < 0.01$; ***$p < 0.001$ (Student's $t$ tests). All $t$ tests were conducted comparing to control cells

activities and arrangements of organelles distributed strategically in the cytoplasm[1,2]. The specific molecules involved in these organelle dynamics and their impact on normal and pathological cellular functions are still little understood. The level of complexity added by organelle interplay requires intense analysis to fully understand cell functioning and plasticity during healthy homeostatic adjustments and the alterations that can lead to disease[15]. Our results identified a molecular machinery that promotes interplay, coordination and mutual regulation between membrane trafficking, nutrient sensing and lipid metabolism, converging upon Golgi-dependent secretion. We show that lysosome relocation and concomitant autophagy induction are co-ordinately regulated to maintain membrane trafficking from the Golgi complex. In the framework of control theory[50], the KDELR–Golgi-based signalling cascade represents a regulatory circuit that crucially contributes to the adaptation of exocytic membrane trafficking to cellular demands[51,52]. KDELR through its Gs-PKA pathway acts as a sensor of incoming membrane trafficking to the Golgi[23]. This pathway coordinates a response that involves membrane trafficking[19], the actin cytoskeleton[19] and the elements added here, such as lysosome repositioning, microtubule molecular motor proteins, autophagy, and LD-dependent energy metabolism. Our results reveal a coordination system that can acutely respond to changes in membrane trafficking input from the ER to the Golgi complex. Such coordination can be even further regulated at the transcriptional level. We previously reported that gene expression related to membrane trafficking and energy metabolism responds positively to activation of KDELR signalling, most likely involving cAMP/PKA-responsive transcription factors such as CREB and ATF1[19]. Here, we add to KDELR function the modulation of TIDeRS as an activity- and location-dependent machinery that coordinates membrane trafficking and autophagy-derived nutritional supplies. Indeed, understanding how membrane trafficking, autophagy and nutrient sensing are spatiotemporally coordinated for the maintenance of cell behaviour and homeostasis might be helpful for the development of new therapeutic tools, especially considering KDELR as a pharmacological target.

## Methods

**Reagents**. 8-Br-cAMP was from Tocris Biosciences. DD-solubilizer was from Clontech (ARIAD Pharmaceuticals Inc., Cambridge, MA, USA). TransIT-LT1 transfection reagent was from Mirus Bio LLC. Lipofectamine 2000 transfection reagent and BODIPY, 4,4-difluoro-5-(2-thienyl)-4-bora-3$a$,4$a$-diaza-$s$-indacene-3-propionic acid, succinimidyl ester (558/568 nm) and 6-(((4,4-difluoro-5-(2-thienyl)-4-bora-3$a$,4$a$-diaza-$s$-indacene-3-yl)styryloxy)acetyl)aminohexanoic acid, succinimidyl ester (630/650 nm) were from Life Science Invitrogen Corporation.

RNeasy mini kits were from Qiagen GmbH. Octa-arginine cell-permeant peptides, R8-Gs (RRRRRRRR-RVFNDCRDIIQRMHLRQYELL) and R8-PKI (RRRRRRRR-GRTGRRNAI) were synthesized by Dr. Guzman (Núcleo Biotecnología Curauma, Pontificia Universidad Católica de Valparaíso, Valparaíso, Chile). Cell-permeable myristoylated-PKI peptide was from Calbiochem (Merck-Millipore). The CFFKDEL (KDEL) and CFFKDEA (KDEA) peptides were synthesized and conjugated to BODIPY (Invitrogen) by Primm Srl (Milan, Italy). Lysotracker was from Molecular Probes (Thermo Fisher).

**shRNAs and plasmids**. KDELR1-GFP, KDELR2-GFP, KDELR3-GFP, KDELR-D193N-GFP, Gs and Gq minigenes and ADCY9-sh were described elsewhere[19]. LC3-GFP, LC3-GFP-mcherry and p62/SQSTM1-mcherry were provided by Terje Johansen (Department of Medical Biology, The Arctic University of Norway, Norway). pGIPzdynC1I1, pGIPzdynC1I2, pGIPzdynLRB1, pGIPzdynLL1 and pGIPzdynLL2 (Open Biosystems) were used to deliver shRNAs. pCMV-HA-DynLRB1 was generated by cloning the DynLRB1 coding sequence from complementary DNA of 293T cells as previously described[53]. Site-directed mutagenesis on DynLRB1 or p62/SQSTM1-mcherry was performed using Quick-Change Lightning Site (Agilent Technologies) with pCMV-HA-DynLRB1 or pCMV-p62/SQSTM1-mcherry, respectively, used as templates. VSVG-GFP and LDLR-Y18A-GFP[54], RUSH-GalT-GFP, RUSH-EGFR and RUSH-LAMP1-GFP were described elsewhere[29]. The generation of ATG5-silenced cells was performed as described elsewhere[55].

**Antibodies**. The anti-p62/SQSTM1 (Cat# 814802, dilution IFI 1/100; WB 1/1000 (v/v)) antibody was from BioLegend, the anti-LC3-B (Cat# 12741, dilution IFI 1/100 (v/v); WB 1/1000 (v/v)) and ATG5 (Cat# 12994, dilution IFI 1/100 (v/v); WB 1/1000 (v/v)) antibodies were from Cell Signaling Technology and the anti-HA (Cat# MMS-101R, dilution IFI 1/100 (v/v); WB 1/1000 (v/v), now available from BioLegend) antibody was from Covance. The anti-GFP (Cat# 029762, dilution WB 1/1000 (v/v)) antibody was from Chromotech. The anti-β-actin (Cat# PA1–183, dilution IFI 1/1000 (v/v)) antibody was from Thermo Fisher Scientific. The anti-mcherry (Cat# ab167453, dilution WB 1/5000 (v/v)) and anti-tubulin (Cat# ab18207, dilution WB 1/5000 v/v) antibodies were from Abcam.

**Cell lines**. HeLa or H4 cells were grown in Dulbecco's modified Eagle's medium (DMEM) supplemented with 10% foetal calf serum. HeLa-expressing hGH-FM-GFP cells[27] were grown in DMEM supplemented with 10% foetal calf serum in the absence of DD-solubilizer. HeLa cells were transfected with plasmid vectors using TransIT-LT1, following the manufacturer's protocol. All cell lines were obtained from ATCC.

**Cell microinjection**. HeLa cells were microinjected as described previously[54] using a semiautomatic microinjector (Femtojet and InjectMan NI2; Eppendorf).

**Synchronized ER-to-Golgi traffic**. The synchronized expression of LAMP1-GFP, albumin-GFP or furin-GFP was performed by microinjection of the respective DNA constructs, and traffic pulses from the ER were performed as reported previously[19]. Synchronized release from ER using the RUSH system was performed as reported[29]. Synchronized hGH-FM-GFP release from the ER was performed by the addition of DD-solubilizer throughout the experiment as described previously[19,27]. Some experiments were performed using a Leica DMI 6000 microscope (×63 water immersion objective, EMCCD Andor Xion camera) running the LASF Leica software, but mostly by using a TCS SP8 spectral confocal microscopy (×63 oil immersion objective, 1.4 N.A.) running the LASX Leica software.

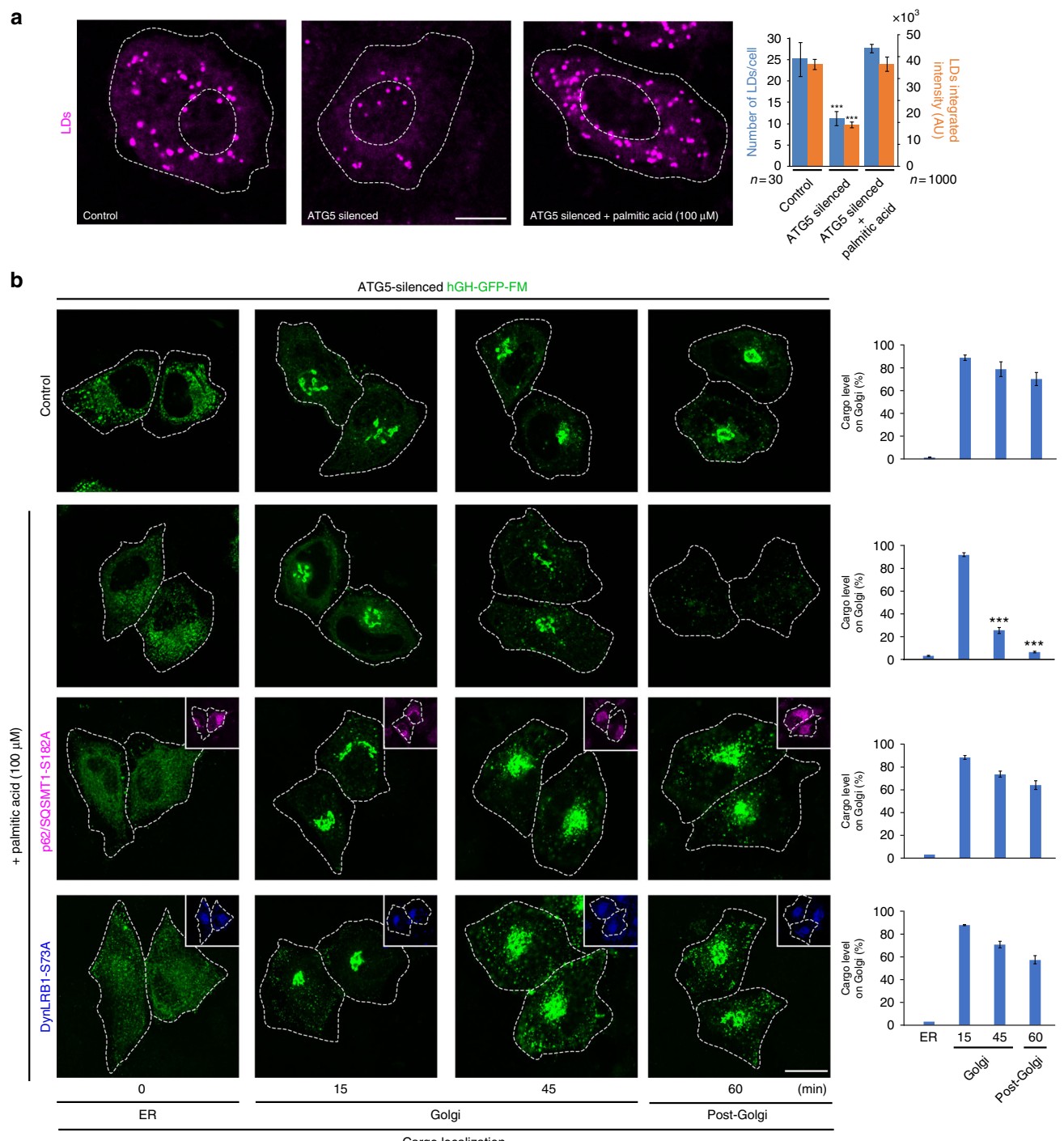

**Fig. 9** Autophagy-dependent lipid droplet turnover affects protein secretion. **a** HeLa cells expressing human growth hormone fused to the polymerization/depolymerization FM domain (hGH-GFP-FM) were left untreated (Control) or subjected to ATG5 silencing (ATG5 silenced) and kept in normal media (Control and ATG5 silenced) or incubated for 4h in media supplemented with 100 µM palmitic acid (third image), followed by lipid droplet (LD) staining with BODIPY. Scale bar, 10 µm. The graph shows the quantification of the number and fluorescence intensity of LDs ($n = 1000$ LDs) of cells ($n = 30$ cells) subjected to the indicated treatments, from images as those shown at the left. **b** HeLa cells expressing hGH-GFP-FM were subjected to ATG5 silencing and left without further transfection (upper two panels) or transfected to co-express either p62/SQSMT1-mcherry-S182A or DynLRB1-HA-S73A. Cells were kept in normal media (Control, first panel) or incubated for 4 h in media supplemented with 100 µM palmitic acid (lower three panels), followed by the ER-to-Golgi transport assay. Cells were fixed (upper three panels) or fixed and permeabilized, followed by immuno-staining with antibody to detect DynLRB1-HA-S73A. Representative images show hGH-GFP-FM localization in the endoplasmic reticulum (ER), Golgi or post-Golgi at the indicated times after addition of DD-solubilizer. The respective graph in each panel depicts the quantification of hGH-GFP-FM in the Golgi complex at the times indicated in images as those shown at the left ($n = 30$ cells). Scale bar, 10 µm. Data are means ± SEM. ***$p < 0.001$ (Student's $t$ tests). All $t$ tests were conducted comparing to control cells. HA hemagglutinin

**Image analysis and processing.** All images for quantitative fluorescence analysis were used as originals without any processing or adjustment. The ImageJ (Fiji distribution) and MetaMorph (Molecular Devices) software were used for quantitative image analysis. The number of autolysosomes, lysosomes, autophagosomes and lipid droplets were semi-automated quantified using the TrackMate plugin on ImageJ. Co-localization and integrated fluorescence analysis were performed using Metamorph, as described previously[54]. For figure elaboration only, the channels of each image were separated, and the level of signal of each channel adjusted separately to facilitate the observation of dim structures. In a few cases, a Sharpen or Gaussian-blur filter was used (radius, 0.2–0.4 pixels) using Adobe Photoshop CS6 (Adobe System Incorporated). All figures were prepared on Illustrator CC (Adobe System Incorporated). Perinuclear localization percentage was calculated from data obtained by using the Radial Profile Angle plugin of the ImageJ software.

**Lysosome perinuclear localization.** Radial integrated fluorescence intensity was measured by using 70 concentric circles centred on the Golgi area, thus covering 15 μm radial distribution of individual cells. The sum of fluorescence integrated intensity of the inner 35 circles were considered as perinuclear and the sum of the outer 36 circles as the cytoplasm. Then, percentage of perinuclear localization was calculated by dividing the sum of perinuclear fluorescence by total fluorescence ([perinuclear/[perinuclear + cytoplasm]] ×100). Each treatment was evaluated by quantifying 10 cells per condition and per experiment.

**p62/SQSTM1 and LDs interaction and speed of joint movement.** Dual colour live-cell imaging was performed on a TCS SP8 Leica confocal microscope. ICY software (Quantitative Image Analysis Unit, Institut Pasteur) was used to perform tracking, interaction and speed analyses of p62/SQSTM1-mcherry and LDs. Multi-channel images were imported to ICY and were pseudo-binary threshold-adjusted for subsequent analysis by the spot detector plugin (configuring the size parameters, detector type and dimensional filters). The resulting threshold-adjusted images were analysed using the spot tracking plugin to establish the movement of the detected fluorescent signals (configuring the tracking parameters) and exported to the track manager, where speeds and interactions were obtained through the Track Processor Instant Speed and Track Processor Interaction Analysis. Interaction was defined as vicinity (three pixels) of red/green signal for at least three frames (30 sec).

**FRAP measurements of p62/SQSTM1 dynamics.** HeLa cells were transfected to express p62/SQSTM1-wt, p62/SQSTM1-S182A or p62/SQSTM1-S182E. After 48 h, cells were placed on an Okolab temperature control chamber at 37 °C on phenol red-free DMEM-HEPES, 5% foetal bovine serum. Fluorescence recovery after photobleaching (FRAP) analysis was performed using the FRAP wizard module and the LASX software (Leica) as previously done in Cancino et al.[19] using zoom-in and zero-background on the Leica SP8, with a ×63 oil immersion objective (1.4 NA), and the 1 AU pinhole. Bleaching was performed using FRAP booster with a 561-nm laser irradiation at 100% power, on individual p62/SQSTM1 puncta. One iteration of 0.5 s was used to achieve 90 to 95% of bleaching. Image acquisition for recovery was every 10 s at 700 Hz scanner speed. FRAP analysis was performed with the FRAP wizard module and the LASX software (Leica), and the recovery time was adjusted to one-component exponential curve fitting and calculated for ten p62/SQSTM1 puncta.

**Preparation and uptake of membrane-permeable peptides.** The labelling of CFFKDEL (KDEL) and CFFKDEA (KDEA) peptides with BODIPY succinimidyl ester was performed as previously described[20]. Cells were treated with the indicated concentrations of peptides for the indicated times, in phenol red-free DMEM-HEPES medium. The 630/650 nm BODIPY of the KDEL fusion peptide was used for different experiments. Octa-arginine (R8) peptides were used at the indicated concentrations and for the indicated times, in phenol red-free DMEM-HEPES medium.

**Lipid-droplet imaging.** Lipid droplets were stained on live or fixed cells by using BODIPY™ 493/503 (4,4-difluoro-1,3,5,7,8-pentamethyl-4-bora-3a,4a-diaza-s-indacene) using the manufacturer's protocol (Invitrogen). Lipid droplets were imaged by 488-nm excitation on a TCS SP8 spectral confocal microscope (×63 oil immersion objective, 1.4 NA) running the LASX Leica software.

**IL-2 secretion.** Mouse LMR7.5 T hybridoma T-lymphocytes were transiently transfected to express p62/SQSTM1-wt or p62/SQSTM1-S182A (both fused to mcherry) for 16 h. Then, cells were activated during 3 h in co-culture with antigen-loaded B-lymphocytes as previously described[56]. The culture medium from each condition was obtained at different time points, and IL-2 was measured using the Opt EIA kit as indicated in the manufacturer's protocol (BD Bioscience).

**Immunoprecipitation of p62/SQSTM1 and DynLRB1.** HeLa cells were transfected to express p62/SQSTM1-mcherry (or its mutants) alone or in combination with LC3-GFP or DynLRB1-HA. Cell lysates were prepared using lysis buffer (50 mM HEPES, pH 7.4, 100 mM NaCl, 0.5% NP-40, 5 mM EDTA, 5 mM EGTA, and a cocktail of protease inhibitors from Roche). Immunoprecipitation using the indicated antibodies bound to magnetic beads was performed as previously described[19].

**Electron microscopy, data acquisition and quantification.** Samples were prepared as described previously[57]. Briefly, cells were fixed with 1% glutaraldehyde in 0.15 M HEPES for 2 h at room temperature. Cells were stained with $OsO_4$/potassium ferrocyanide, dehydrated and Epon-embedded, as described previously[57]. For immune-nanogold, cells were grown in six-well plates, fixed with 4% paraformaldehyde and 0.05% glutaraldehyde, and permeabilised with 0.2% saponin in blocking solution (phosphate-buffered saline containing 1% bovine serum albumin, 50 mM $NH_4Cl$). Cells were then incubated with antibodies to the antigen of interest overnight at 4 °C, followed by incubation for 2 h at room temperature with secondary antibodies labelled with nanogold. Gold particles enhancing was by nanogold enhancer (Nanoprobes), according to the manufacturer's instructions. Subsequently, samples were Epon-embedded, and ultra-thin sections (50–70 nm) were obtained and examined by electron microscopy (Tecnai-12; FEI, Eindhoven). Images were acquired using a CCD digital camera (Veletta, FEI, Eindhoven, The Netherlands). To score lysosome distribution, random sampling of Golgi profiles was carried out, and the number of endo-lysosomal structures surrounding the Golgi area within 1.5 μm was estimated. Endo-lysosomal structures were defined as membrane-bound structures with more than eight intraluminal vesicles or other amorphous electron-dense material.

**Public databases.**

1. NetPhos 3.1 (http://www.cbs.dtu.dk/services/NetPhos/).
2. iPTMnet (http://research.bioinformatics.udel.edu/iptmnet/).
3. PhosphoSitePlus (http://www.phosphosite.org/homeAction.action).

## Data availability

All data generated or analysed during this study are included in this published article (and its supplementary information files).

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

## Acknowledgements

We would like to thank all colleagues at the Centro de Biología Celular y Biomedicina (CEBICEM, Chile) and the Istituto di Biochimica delle Proteine (IBP, Italy) who kindly provided reagents and technical support, Dr. A. Peden (Department of Biomedical Science, University of Sheffield, Western Bank Sheffield, UK) for hGH-FM-GFP-expressing HeLa cells. J.C. was supported by FONDECYT No. 1150986 and UNAB DI-646–15R, G.A. by FONDECYT No. 1130852, M.I.Y. by FONDECYT 1141182, G.A.M. by FONDECYT No. 1161252, P.V.B. by FONDECYT No. 1171649, R.R. by FIRC Fellowship 15111, A.G. by FONDECYT No. 1181907 and CONICYT Basal Programme No. AFB-170005. V.A.C. was supported by CONICYT fellowship 21151194.

## Author contributions

D.T., J.E., T.J., C.Z., R.R., D.F., V.A.C., A.S. and A.G.-C. carried out the experiments. S.H. and G.A.M. quantitatively analysed fluorescent images. F.G. synthesized membrane-permeable peptides. G.A. prepared shRNAs and DynLRB1 mutants. M.I.Y. and P.V.B. designed the biochemical experiments, A.G., A.L. and G.A.M. co-wrote the manuscript, and J.C. designed the experiments, directed the project and co-wrote the manuscript.

## Additional information

**Competing interests:** The authors declare no competing interests.

