## [Peer Review File · Nature Communications]

Reviewers' comments:

Reviewer #1 (Remarks to the Author):

In this work, Tapia and colleagues analyze the impact of the distribution of the lysosomes and autophagy on the secretory pathway. They first show the juxta nuclear redistribution of the lysosomes when synchronized cargo reach the Golgi. Then, they show that lysosomes redistribution in the Golgi area depends on the KDEL receptor activation that phosphorylates the dynein light chain DynLRB1 and thus could likely regulate the displacement of the lysosomes from the cell periphery to the cell center. In addition, they report that the juxta nuclear distribution of the lysosomes induced by the cargo trafficking enhanced autophagy. Overall this is an interesting work that support a new mechanism coordinating the secretory and the degradative pathways. The experiments are well performed. After addressing my comments below regarding the requirement of additional controls, the presentation of the data and the English, I think that the manuscript will fulfil the conditions to be published in Nature communication.

- 1) Figures 1b and 2a: EM images are too small. It is difficult to recognize some ER on the figure 1a and to evaluate the difference regarding the accumulation of MVB nearby the Golgi after KDEL-bodipy treatment figure 2a.
- 2) Figure S3a: How long the cells have been treated with the drugs? What is the treatment of the two columns next to those for the control?
- 3) Figure 3C: the difference between the two mutants can also reflect a difference in their expression level. It is important to show the quantification of the expression level of the two mutants by western blot
- 4) Figure 3D: Lamp1-mCherry is already concentrated near the Golgi in the control cells although it should be dispersed in the cytoplasm when the cargo is in the ER.
- 5) Figure S4: The distribution of the lysotracker cannot be visualized. The insert is much too small.
- 6) Figure legend of 5b-5c is not clear.
- 7) Figure 5D: the line along which the fluorescence intensity has been measured in the insert should be drawn on the cell.
- 9) Lines 138-143 and line 263: the description of data not shown should be eliminated
- 8) The discussion lines 333-344 is very speculative and could be skipped.
- 9) The number of cell analyzed in each experiment as well as the number of experiment (3 experiments should be performed for each piece data) for which the authors perform some quantification are missing in the figure legends. Furthermore, it is important to precise which data are compared in the T tests. This can be marked on the figures by a line between the two columns analyzed.
- 10) English should be improved, typing, spelling and grammar errors should be corrected eg lines 33, 80, 101, 321

Reviewer #2 (Remarks to the Author):

Nat Comms review

Lysosomal redistribution and autophagy induction regulate golgi-mediated secretion in response to KDEL activation during ER-to-Golgi trafficking

The manuscript by Tapia et al describes their findings that inter-organelle communication is supported by protein and organelle re-localisation and cellular signalling to control protein release from Golgi and secretion at the plasma membrane. This concept is very interesting and as the authors discuss, it is an area of cell biology that is little explored and warrants further investigation. The authors are extending work previously published and they have attempted to comprehensively study the signalling pathway controlling protein exit from Golgi and secretion.

Having said that, I don't feel that the data included in the current manuscript is of sufficient robustness and depth to be considered for Nature Communications at this point. I think the authors make claims that the experiments do indeed indicate but, at least for this reviewer, are more correlative and not proven and in many cases, the paper lacks the necessary controls to support the authors conclusions. Another general, major comment is that there is no consistency Figure to Figure how the authors choose to activate the pathway (e.g. by overexpression of KDELR vs KDEL ligand), how it is inhibited (R8-Gs vs shRNA for KDELR) and the readout (lysosomal localisation, hGH-GFP exit from Golgi by IF, secretion of hGH-GFP into the media). These discrepancies make it extremely difficult to interpret the data as a whole and I believe all of the combinations described above should be used to address every stage of the paper. In particular I think that the role for PKA signalling is not well developed and Figures 5 and 6 are weak, demonstrate the same point and could be combined. For example, what is the difference between Figure 6c-d and S5a except the addition of lysotracker? The graphs seem to show quantification of exactly the same thing? As far as this reviewer is concerned, there is no evidence that selective autophagy is involved in this process and I also do not think that the role of lysosomal localisation vs autophagy activation is clear (i.e. cause vs consequence). As I mentioned before however, I think the study is very interesting and I would thoroughly encourage the authors to increase the robustness and rigour of their experiments after which I believe it would be an interesting addition to the field.

General comments

- The title doesn't necessarily reflect the conclusions of the paper. All of the data indicates that lysosomal localisation/autophagy activation support exit of proteins from the Golgi not necessarily ER-Golgi transport.
- The summary indicates that the pathway described here is required for cell homeostasis but no experiments have been included to support this conclusion.
- Is lysosomal re-localisation specific to KDEL and its receptor? Are any other ER-Golgi exit/secretion systems controlled by the same lysosomal relocalisation and autophagy induction?
- There is a lot of 'data not shown'. This data should all be included as a major criticism of this paper is the lack of a control experiments.
- Figure 1 requires a vehicle only control and solubilizer alone to confirm changes in lysosome localisation are not a result of i.e. changes in intracellular pH. The same controls need to be included for example when looking at PKA signalling and autophagy induction.
- The EM in Figure 2 and Figure 3 are quite different. For example in Figure 2, they contain a lot of vesicular structures and are thus more reminiscent of MVB or autophagosomes. In Figure 3 however, they are largely empty vesicular structures. Furthermore in Figure 2, there simply seems to be more lysosomes, not that they are necessarily closer to the labelled Golgi. Can the author reconcile these differences?
- There is no reference in the text as to why over-expression of KDELR1 alone is sufficient to cause the redistribution of lysosomes. Can the authors provide evidence that this overexpression increases signalling?
- The paper would benefit from including evidence that the phospho-mimetic and phospho-inert mutations of DynLRB1 and p62 function as the authors suggest. Could any of these mutations affect the protein structure?
- Please include evidence by Western blot of all knock-down/knock-outs.
- Are all experiments involving protein exit from Golgi carried out in the presence of solubilizer for the same amount of time? Please include this information in all the legends.
- What is the link between antigen presentation in T-cells and KDELR activation/signalling? And why was p62 used in Figure 6e rather than for example Atg5 KD which are a very strong effect on protein secretion.
- In Figure S5, the authors suggest there might be decreased fusion of autophagosomes (p62-positive structures) and lysosomes and thus mutant p62 levels increase rather than being degraded. However there is no indication of why this might be. Does p62 affect lysosomal localisation or lysosomal function? Does inhibiting lysosomal function have any further effect i.e. look at p62 and autophagic flux.

Minor comments

- The summary is very dense and not very accessible to someone who is not an expert in this specific KDEL receptor pathway.
- Check reference to figure S3 in the text.
- There are quite a few spelling/grammatical errors in the text. Also a spelling error in Figure 1a (Lmap1 instead of Lamp1).
- There is no mention of cytochalasin B (Figure S3a) in the text.
- Please check all of the graph axis labels for consistency. Why is the axis on Figure S3b, c different to others?
- Please check consistency throughout figures in terms of text size etc
- It is not appropriate to label graph axis 'p62 autophagosomes'. Please change to 'p62-positive structures or similar'.
- Figure 3d does not show a robust relocalisation of lysosomes.
- The graph in Figure S4a is different to the other graphs showing similar experiments. For consistency, please change
- In Figure 5d, please include evidence that DynLRB transfection
- Figure 6b would benefit from splitting the channels, it is not easy to see the red vs blue punctate structures.
- Change the axis on graphs in Figure 5a,b,c so they are consistent.
- What does 'autophagic' mean in Figure 5A

Reviewer #3 (Remarks to the Author):

The manuscript by Tapia et al reports on the identification of a link between autophagy and exit of secretory cargo from the Golgi. This link is mediated via a traffic-induced signaling event. The topic that this paper investigates is important and addresses poorly understood areas of cell biology, namely the crosstalk of secretion and autophagy as well as the autoregulation of the secretory pathway. Although the basic principle of this work is very interesting, the paper suffers from a lack of many controls and mechanistic insight. For instance, the work is based on the analysis of a single cargo, but yet the authors claim to have identified a universal cellular pathway, which they call TIDERS. In my opinion, this cannot be concluded by the analysis of a single cargo, using a single secretion system in a single cell line. Moreover, there is very little mechanistic insight into the role of the traffic-induced signaling event (PKA activation) and the regulation of secretion and autophagy. I think that this work is highly interesting and therefore I recommend that the authors invest a substantial amount of further work to revise this paper for a journal like Nature Communications. Below are my specific comments, which are ordered according to figures:

1- In Figure 1: the analysis of lysosomal positioning is not accurate, because it only tells how many cells in the population show a certain phenomenon. It would be good to complement this with an analysis of the fluorescence intensity of lysosomes in the Golgi-region versus outside the Golgi region.

2- As stated above, it would be necessary to complement the analysis in this paper with another trafficking assays using another type of cargo. My recommendation is to use the RUSH system, which the authors are certainly aware of. Here the authors may use two different types of cargo (e.g. single pass transmembrane, multipass transmembrane, soluble, GPI-anchored, etc..) to show that traffic-induced lysosomal redistribution is a general response. Other trafficking assays exist such as temp-sensitive VSVG, but this is in my opinion outdated as it suffers from the usage of different culturing temperatures. Secretion of collagen might also be used in case the authors want to show the effect of bulky cargo.

3- I might have missed it, but Figure 2 lacks several controls. No images of cells transfected with control siRNA are shown. In addition, the analysis of cell number should be complemented with the analysis of fluorescence intensity as recommended in point 1.

4- In Figure 2: I do not understand why KDEL-BODIPY has no effect on cell expressing KDEL3. Don't these cells have endogenous KDEL1, which might be activated. This is not commented on by the authors and the result as such does not make much sense.

5- Figure 3 suffers again from the fact that the whole work is based on the analysis of a single cargo using a single secretion system. I strongly recommend to complement this analysis with the RUSH assay.

6- the results shown in Figure 4 are confusing. The inhibition of autophagy using a chemical inhibitor does not prevent lysosomal repositioning. Therefore, the localization of lysosomes and post-Golgi trafficking are not linked to each other. In addition, the basal level of autophagy is pretty high, but the authors state that this is not the case. Nevertheless, the levels of LC3-II in panel b disagree with the text. Furthermore, it seems that it is basal autophagy (i.e. not the starvation induced) that regulates Golgi exit. Conversely, arrival of a cargo wave to the Golgi activates autophagy. Thus, it might be that the high basal levels of autophagy in these cells might be explained by the fact that they have an overloaded ER (ER overload → UPR → upregulation of secretory capacity → more cargo arriving at Golgi → activation of autophagy). This is just one possible scenario, and I am certain more can be thought of. This shows that the level of analysis conducted here requires some further depth.

7- In Figure 6, it is necessary to show that p62 is phosphorylated by PKA (using an in vitro kinase assay) and that it is phosphorylated when there is a cargo wave arriving at the Golgi. This can be either done with phosphoproteomics, or by IP of p62 and determining its phosphorylation by immunoblotting.

8- The role of autophagy is unclear in figures 4-6. It clearly seems that autophagy plays a role in exit from the Golgi. However, instead of providing mechanistic insight, the authors move on and show that selective autophagy (p62-dependent) plays a role. Again it is not clear why type of selective autophagy is involved. Overall, I think that the analysis of the role of autophagy is a bit preliminary and requires more mechanistic depth.

Reviewers' comments

Reviewer #1 (Remarks to the Author):

In this work, Tapia and colleagues analyze the impact of the distribution of the lysosomes and autophagy on the secretory pathway. They first show the juxta nuclear redistribution of the lysosomes when synchronized cargo reach the Golgi. Then, they show that lysosomes redistribution in the Golgi area depends on the KDEL receptor activation that phosphorylates the dynein light chain DynLRB1 and thus could likely regulate the displacement of the lysosomes from the cell periphery to the cell center. In addition, they report that the juxta nuclear distribution of the lysosomes induced by the cargo trafficking enhanced autophagy.

Overall this is an interesting work that support a new mechanism coordinating the secretory and the degradative pathways. The experiments are well performed. After addressing my comments below regarding the requirement of additional controls, the presentation of the data and the English, I think that the manuscript will fulfil the conditions to be published in *Nature Communication*.

Our reply: We thank the Reviewer for this encouraging comment. We have done our best to answer his concerns and criticisms as indicated below in our point-by-point replies. We hope the Reviewer will find this revised version really improved and suitable for publication in *Nature Communications*.

1) Figures 1b and 2a: EM images are too small. It is difficult to recognize some ER on the figure 1a and to evaluate the difference regarding the accumulation of MVB nearby the Golgi after KDEL-bodipy treatment figure 2a.

Our reply: We have increased the sizes of EM images.

2) Figure S3a: How long the cells have been treated with the drugs? What is the treatment of the two columns next to those for the control?

Our reply: We pretreated the cells for 15 min and then maintained the drugs for 30 min in the presence of KDELR activator, KDEL-BODIPY. In the graph, the referred two columns next to the control column correspond to KDEL-BODIPY alone, in the absence of other treatments. Therefore, we changed the figure to avoid confusion, which is now figure S2.

3) Figure 3C: the difference between the two mutants can also reflect a difference in their expression level. It is important to show the quantification of the expression level of the two mutants by western blot

Our reply: We consistently see this effect by immunofluorescence imaging. As requested, we measured this by Western blotting with an anti-HA antibody. The levels of expression were similar and were included as normalized level of expression using actin as loading control.

4) Figure 3D: Lamp1-mCherry is already concentrated near the Golgi in the control cells although it should be dispersed in the cytoplasm when the cargo is in the ER.

Our reply: As DynLRB1 expression cannot be visualised directly, we co-expressed the DynLRB1 wt or S73A versions with Lamp1-mCherry to select the cells for cell imaging. Under this condition, the distribution of overexpressed Lamp1 near the perinuclear area is expected. We calculated the radial localisation percentage to measure lysosomal relocalisation. Although in the figure the lysosomes are hardly seen to avoid the saturation of the perinuclear fluorescence, we quantified their distribution.

5) Figure S4: The distribution of the lysotracker cannot be visualized. The insert is much too small.

Our reply: The only reason of the insert is to show that the cell depicted express soluble GFP reflecting their transfection with the shRNA (See figure legend).

6) Figure legend of 5b-5c is not clear.

Our reply: We clarified the Figure legend.

7) Figure 5D: the line along which the fluorescence intensity has been measured in the insert should be drawn on the cell.

Our reply: We included the line for the line scan over the selected structures.

9) Lines 138-143 and line 263: the description of data not shown should be eliminated

Our reply: We now include these data in the main text and figures.

8) The discussion lines 333-344 is very speculative and could be skipped.

Our reply: Our new data give further support to the Discussion. We modified the text accordingly to include the results of lipid droplets.

9) The number of cell analyzed in each experiment as well as the number of experiment (3 experiments should be performed for each piece data) for which the authors perform some quantification are missing in the figure legends. Furthermore, it is important to precise which data are compared in the T tests. This can be marked on the figures by a line between the two columns analyzed.

Our reply: We included the numbers of cells for each experiment and the numbers of experiments in the figure legend. T-tests always compare the treated conditions against the control, otherwise is indicated on the figure.

10) English should be improved, typing, spelling and grammar errors should be corrected eg lines 33, 80, 101, 321

Our reply: The full manuscript has been edited by a professional scientific English editor. We hope it is now suitable for publication.

Reviewer #2 (Remarks to the Author):

Lysosomal redistribution and autophagy induction regulate golgi-mediated secretion in response to KDEL activation during ER-to-Golgi trafficking

The manuscript by Tapia et al describes their findings that inter-organelle communication is supported by protein and organelle re-localisation and cellular signalling to control protein release from Golgi and secretion at the plasma membrane. This concept is very interesting and as the authors discuss, it is an area of cell biology that is little explored and warrants further investigation. The authors are extending work previously published and they have attempted to comprehensively study the signalling pathway controlling protein exit from Golgi and secretion.

Having said that, I don't feel that the data included in the current manuscript is of sufficient robustness and depth to be considered for *Nature Communications* at this point. I think the

authors make claims that the experiments do indeed indicate but, at least for this reviewer, are more correlative and not proven and in many cases, the paper lacks the necessary controls to support the authors' conclusions. Another general, major comment is that there is no consistency Figure to Figure how the authors choose to activate the pathway (e.g. by overexpression of KDELR vs KDEL ligand), how it is inhibited (R8-Gs vs shRNA for KDELR) and the readout (lysosomal localisation, hGH-GFP exit from Golgi by IF, secretion of hGH-GFP into the media). These discrepancies make it extremely difficult to interpret the data as a whole and I believe all of the combinations described above should be used to address every stage of the paper.

In particular I think that the role for PKA signalling is not well developed and Figures 5 and 6 are weak, demonstrate the same point and could be combined. For example, what is the difference between Figure 6c-d and S5a except the addition of lysotracker? The graphs seem to show quantification of exactly the same thing? As far as this reviewer is concerned, there is no evidence that selective autophagy is involved in this process and I also do not think that the role of lysosomal localisation vs autophagy activation is clear (i.e. cause vs consequence). As I mentioned before however, I think the study is very interesting and I would thoroughly encourage the authors to increase the robustness and rigour of their experiments after which I believe it would be an interesting addition to the field.

Our reply: We think these general comments might reflect some misunderstanding of the experiments perhaps due to a confused text, which we tried to improve all over the manuscript. We cannot agree with the lack of consistency mentioned by the Reviewer.

As detailed in the manuscript, lysosome relocation was initially seen as an unexpected response for ER-to-Golgi membrane input, a cellular process that activates KDELR signalling (Cancino et al. Dev Cell 2014). In the present work, we show similar lysosome relocation when overexpressing one of the KDELR isoforms, while the others had no effect. It is known that overexpression of GPCRs in general determine their activation. We also used KDEL-ligand to directly activate the KDELR with similar effects. We think reasonable to conclude that KDELR activation leads to lysosomal relocation. Furthermore, as control, we silenced the KDELR1 isoform whose overexpression relocated the lysosomes and show no effect of the KDEL-ligand under this condition. We designed each experiment to evaluate, piece by piece, the involvement of KDELR and its downstream pathway leading to lysosome relocation. All the results are consistent with the conclusion that KDELR signalling towards the PKA pathway accounts for the lysosomal relocation. The other readouts mentioned by this reviewer were selected to evaluate the effects on secretion of the components identified as regulators of lysosome repositioning (meaning DynLRB1 and p62). For example, secretion of hGH-GFP into the medium assessed by immunoblot and hGH-GFP exit from Golgi assessed by immunofluorescence are two complementary readouts. The first shows biochemically that the cargo is retained and the second shows where the cargo was retained. Additionally, we selected a battery of possibilities to inhibit the KDELR downstream pathway and test the effect on lysosome relocation and Golgi-dependent secretion. We inhibited KDELR signalling by KDELR1 silencing, a KDELR1 dominant-negative version, a R8-Gs peptide previously shown to block Gs activation, adenylyl cyclase ADCY9 silencing and PKI peptide. All the results point to a role of Gs-mediated cAMP/PKA signalling. In general, this was the current approach in our study. The use of different treatments to interfere with the process at different levels gives strength to the conclusions.

General comments:

- **The title doesn't necessarily reflect the conclusions of the paper. All of the data indicates that lysosomal localisation/ autophagy activation support exit of proteins from the Golgi not necessarily ER-Golgi transport.**

Our reply: We changed the title to include the major conclusions regarding lysosome relocation and lipid droplet degradation. On the other hand, we show that lysosome repositioning is triggered by ER-to-Golgi transport. So far, we have not shown that lysosome repositioning supports ER-to-Golgi transport.

- **The summary indicates that the pathway described here is required for cell homeostasis but no experiments have been included to support this conclusion.**

Our reply: We removed this phrase. However, we would like to remark that membrane transport and autophagy are well accepted as important processes in cell homeostasis.

- **Is lysosomal re-localisation specific to KDEL and its receptor? Are any other ER-Golgi exit/secretion systems controlled by the same lysosomal relocalisation and autophagy induction?**

Our reply: Perhaps there is some confusion. This is the first description of how ER-to-Golgi input can impact on lysosome repositioning through KDEL signalling.

- **There is a lot of ‘data not shown’. This data should all be included as a major criticism of this paper is the lack of a control experiments.**

Our reply: We have included these data in the main text and Figures.

- **Figure 1 requires a vehicle only control and solubilizer alone to confirm changes in lysosome localisation are not a result of i.e. changes in intracellular pH. The same controls need to be included for example when looking at PKA signalling and autophagy induction.**

Our reply: DD-Solubiliser alone had no effects on cells lacking hGH-GFP-FM as cargo. Therefore, we can discard changes on intracellular pH as the main driver of the observed effects.

- **The EM in Figure 2 and Figure 3 are quite different. For example in Figure 2, they contain a lot of vesicular structures and are thus more reminiscent of MVB or autophagosomes. In Figure 3 however, they are largely empty vesicular structures. Furthermore in Figure 2, there simply seems to be more lysosomes, not that they are necessarily closer to the labelled Golgi. Can the author reconcile these differences?**

Our reply: The Reviewer is right; both images are different because in Figure 2 the cells were activated by the KDEL ligand, which triggers the response for lysosome repositioning and autophagy induction (including changes in membrane trafficking). In contrast, in Figure 3, the cells were transfected to express the Dyn mutant in the absence of any KDEL activation. So, no autophagy was induced and membrane transport was not modified. Even more, the EMs for these two Figures are not the same. In Figure 2, the cells were embedded in Epon epoxy resin, whereas in Figure 3 ultra-cryotome was used to cut the sample. So, the preservation of structures in both techniques is not the same, as it is well known. We have included this information on the Figure legend to avoid misinterpretations. Please note that the Figure 3 have been changed to Figure 4.

- **There is no reference in the text as to why over-expression of KDELR1 alone is sufficient to cause the redistribution of lysosomes. Can the authors provide evidence that this overexpression increases signalling?**

Our reply: We now made referenced to a previous paper by Cancino et al. in Dev Cell 2014 showing that overexpression of KDELR1 increases PKA signalling.

- **The paper would benefit from including evidence that the phospho-mimetic and phospho-inert mutations of DynLRB1 and p62 function as the authors suggest. Could any of these mutations affect the protein structure?**

Our reply: We have no data to support changes in protein structure due to point mutations in DynLRB1 or p62/SQSTM1. However, the effect of the point mutations is consistent with the explicative model assuming a role as functionally perturbed proteins.

• Please include evidence by Western blot of all knock-down/knock-outs.

Our reply: For the knock-downs we included just RT-PCR evidence because no suitable antibodies against DynLRB1 are available.

• Are all experiments involving protein exit from Golgi carried out in the presence of solubilizer for the same amount of time? Please include this information in all the legends.

Our reply: We have included this information on the Methods as well as on the Figure and figure legends.

• What is the link between antigen presentation in T-cells and KDEL activation/signalling? And why was p62 used in Figure 6e rather than for example Atg5 KD which are a very strong effect on protein secretion.

Our reply: There is no a particular reason to use p62 instead of ATG5 to perturb secretion. The important point here is that we reproduced the effects of the p62S182A mutant testing an endogenous cargo secretion such as IL2 in a relevant physiological system.

• In Figure S5, the authors suggest there might be decreased fusion of autophagosomes (p62-positive structures) and lysosomes and thus mutant p62 levels increase rather than being degraded. However there is no indication of why this might be. Does p62 affect lysosomal localisation or lysosomal function? Does inhibiting lysosomal function have any further effect i.e. look at p62 and autophagic flux.

Our reply: It has been well documented that p62 is recruited to autophagosomes and that its levels decrease after autophagosomes fuse with lysosomes. In our experiments we show that both p62 wt and its S182A mutant are similarly recruited to autophagosomes, as reflected in a similar increase of their presence in punctate structures. However, while p62 wt co-localises with lysosomes and then its levels decrease, most likely due to lysosomal degradation, p62-S182A show a lower lysosomal co-localization and its punctate structures does not decrease.

Minor comments:

• The summary is very dense and not very accessible to someone who is not an expert in this specific KDEL receptor pathway.

Our reply: We have changed the summary to make it more accessible.

• Check reference to figure S3 in the text.

Our reply: We have checked the reference to Figure S3 on the text.

• There are quite a few spelling/grammatical errors in the text. Also a spelling error in Figure 1a (Lmap1 instead of Lamp1).

Our reply: The full manuscript has been reviewed and edited by a professional scientific English editor.

• There is no mention of cytochalasin B (Figure S3a) in the text.

Our reply: We corrected this missing point.

- **Please check all of the graph axis labels for consistency. Why is the axis on Figure S3b, c different to others?**

Our reply: We have corrected the consistency errors.

- **Please check consistency throughout figures in terms of text size etc**

Our reply: We carefully corrected the consistency errors.

- **It is not appropriate to label graph axis ‘p62 autophagosomes’. Please change to p62-positive structures or similar.**

Our reply: Changed accordingly.

- **Figure 3d does not show a robust relocalisation of lysosomes.**

Our reply: We reproduce here the answer to Reviewer 1, who made a similar comment. As DynLRB1 expression cannot be visualised directly, we co-expressed the DynLRB1 wt or S73A versions with Lamp1-mCherry to select the cells for cell imaging. Under this condition, the distribution of overexpressed Lamp1 near the perinuclear area is expected. We calculated the radial localisation percentage to measure lysosomal relocalisation. Although in the figure the lysosomes are hardly seen to avoid the saturation of the perinuclear fluorescence, we quantified their distribution.

- **The graph in Figure S4a is different to the other graphs showing similar experiments. For consistency, please change**

Our reply: Changed.

- **In Figure 5d, please include evidence that DynLRB transfection**

Our reply: DynLRB1 versions were co-transfected and is now shown on the other figures.

- **Figure 6b would benefit from splitting the channels, it is not easy to see the red vs blue punctate structures.**

Our reply: We have changed Figure 6b to Figure 7b and channels were changed to green and red to improve visualization.

- **Change the axis on graphs in Figure 5a,b,c so they are consistent.**

Our reply: We split Figure 5 and now the data is on Figure 6 and Figure 7.

- **What does ‘autophagic’ mean in Figure 5A**

Our reply: We have changed “autophagic” to “cells with autophagosomes”.

Reviewer #3 (Remarks to the Author):

The manuscript by Tapia et al reports on the identification of a link between autophagy and exit of secretory cargo from the Golgi. This link is mediated via a traffic-induced signaling event. The topic that this paper investigates is important and addresses poorly understood areas of cell biology, namely the crosstalk of secretion and autophagy as well as the autoregulation of the secretory pathway.

Although the basic principle of this work is very interesting, the paper suffers from a lack of many controls and mechanistic insight. For instance, the work is based on the analysis of a single cargo, but yet the authors claim to have identified a universal cellular pathway, which they call TIDERS. In my opinion, this cannot be concluded by the analysis of a single cargo, using a single secretion system in a single cell line. Moreover, there is very little mechanistic insight into the role of the traffic-induced signaling event (PKA activation) and the regulation of secretion and autophagy.

Our replay: In our revised version we now include more controls and experiments providing a mechanistic explanation to the effects of lysosomal relocation and autophagy as critical processes to sustain Golgi-dependent secretion. We show that lipid droplet (LD) degradation that accompanies both lysosomal relocation and autophagy, as interrelated processes, can account for the secretion requirement. The new data explain the effects of ATG5-dependent and p62-dependent maintenance of cargo secretion. We included evidence for LDs turnover mediated by autophagy. Overexpression of KDELR or DynLRB1 both reduced LDs number and content. In contrast, their silencing increases the number of LDs. We also show that p62 and dynLRB1 regulate LD degradation in a KDELR-dependent manner. We found reduced number and content of LDs in ATG5 silenced cells resulting in cargo block at the Golgi complex. Restoration of LDs number and content, by fatty acid supplementation, restored normal cargo transport in these cells. However, the expression of p62 or DynLRB1 phospho-defective mutants abrogated the effects of LD restoration by fatty acids in ATG5-silenced cells. In addition, our results show that LDs turnover, which is dependent on KDELR, p62 and DynLRB1, is modulated by PKA signalling.

Previous studies have shown that KDELR is a Golgi sensor of input membrane traffic independently of cargo. This sensor is constitutive to all kind of cells as it is part of a general mechanism by which KDELR-bearing chaperones return to the ER from the Golgi. Here we used a well-established artificial marker of the secretion pathway, which has been extensively validated as a reporter of the trafficking activity from the ER to the Golgi. However, we also include experiments on T cell secretion IL-2, showing that it is inhibited by a p62 mutant that also inhibited the secretion of the artificial reporter. The universality and evolutionary conserved KDELR function, as well as the secretory route, support the notion that our present observations reflect a general mechanism.

I think that this work is highly interesting and therefore I recommend that the authors invest a substantial amount of further work to revise this paper for a journal like *Nature Communications*.

Our reply: We thank the Reviewer for the positive comments and the encouraging to improving our work. In this revised version we attempted to answer all concerns.

Below are my specific comments, which are ordered according to figures:

1- In Figure 1: the analysis of lysosomal positioning is not accurate, because it only tells how many cells in the population show a certain phenomenon. It would be good to complement this with an analysis of the fluorescence intensity of lysosomes in the Golgi-region versus outside the Golgi region.

Our reply: We included a more accurate quantitative lysosomal repositioning analysis calculated the perinuclear localisation of lysosomes by measuring the radial fluorescence integrated intensity, as detailed in the Methods.

2- As stated above, it would be necessary to complement the analysis in this paper with another trafficking assays using another type of cargo. My recommendation is to use the RUSH system,

which the authors are certainly aware of. Here the authors may use two different types of cargo (e.g. single pass transmembrane, multipass transmembrane, soluble, GPI-anchored, etc..) to show that traffic-induced lysosomal redistribution is a general response. Other trafficking assays exist such as temp-sensitive VSVG, but this is in my opinion outdated as it suffers from the usage of different culturing temperatures. Secretion of collagen might also be used in case the authors want to show the effect of bulky cargo.

Our reply: It is important to remark that we first observed lysosome repositioning during an analysis of the trafficking of a lysosomal protein, which we then extended to other models testing the traffic from the ER to the Golgi. Our results with a model secretion protein and with the secretion of IL2 demonstrate that lysosomal repositioning is a more general phenomena associated with the activation of the KDEL receptor, which is currently triggered by the membrane input to the Golgi carrying KDEL-bearing chaperones from the ER. Indeed, among our initial models we tested the RUSH system for lysosomal and non-lysosomal proteins. However, this approach resulted inconvenient and most frequently led to inconsistent results. The RUSH system used to retain proteins at the ER depends on a KDEL-based hook for ER retention. We have shown previously that overexpression of KDEL-containing proteins on ER activates KDEL receptor signalling (Pulvirenti et al. 2008 Nat Cell Bio, Cancino et al. 2014 Dev Cell). We tried with several RUSH constructs obtaining the same phenotype: lysosomal repositioning in the absence of protein released from ER. This is most likely due to KDEL receptor signalling activation by the KDEL-hook itself, which being in excess seems to be released from the ER. Therefore, we discarded all these results in the first as well as in this revised version of the manuscript. Instead, to avoid problems with overexpression of KDEL-containing proteins we used microinjection for cargo transport synchronisation and the FM-domain-based ER release. It is important to insist in the notion that KDEL receptor signalling is not triggered by cargo itself. We have previously established that KDEL receptor becomes activated by ER-to-Golgi membrane transport independently of the kind of cargo and cell type. KDEL receptor is activated by the KDEL-bearing chaperones that normally recycle between the ER and Golgi carrying different kind of protein cargo during the course of their chaperone-assisted structural folding and assembly.

3- I might have missed it, but Figure 2 lacks several controls. No images of cells transfected with control siRNA are shown. In addition, the analysis of cell number should be complemented with the analysis of fluorescence intensity as recommended in point 1.

Our reply: We have included the percentage of perinuclear localisation of control cells, the image was not included because of restricted space on the figure. The percentage of perinuclear localization is now included in this Figure, as in Figure 1.

4- In Figure 2: I do not understand why KDEL-BODIPY has no effect on cells expressing KDEL3. Don't these cells have endogenous KDEL1, which might be activated. This is not commented on by the authors and the result as such does not make much sense.

Our reply: This comment reflects that the experiments were not well described and thus the Reviewer's concern seems quite reasonable. However, we did not treat KDEL3-overexpressing cells with KDEL-BODIPY. We only treated KDEL-BODIPY cells silenced for KDEL1 and cells expressing the dominant-negative KDEL1-D193N. It is also necessary to clarify that overexpression of KDEL3 does not result in activation of KDEL1, which instead occurs by the homotypic overexpression of KDEL1.

5- Figure 3 suffers again from the fact that the whole work is based on the analysis of a single cargo using a single secretion system. I strongly recommend to complement this analysis with the RUSH assay.

Our reply: We have performed experiments using RUSH as mentioned above but not with consistent results as explained. Therefore, we avoided this approach. Again, it is necessary to remark that the activation of KDEL signalling is independent of the type of cargo. This is because KDEL signalling is not activated by the cargo that arrives to the Golgi. It is activated the many KDEL-bearing chaperones, such as BIP, protein-disulphuro isomerase, glucose-activated chaperone, temperature-activated chaperone, etc) that, while helping the folding of different cargo, escape from ER associated with the different cargos and then recycle back to the ER together with the KDEL (Pulvirenti et al. 2008 NCB). We have previously tested several cargoes in several cell types (Cancino et al. 2014 Dev Cell) and demonstrated that soluble and transmembrane proteins are retained on the Golgi complex when KDEL signalling is blocked (Pulvirenti et al. 2008 NCB, Giannotta et al. 2012 EMBOJ). Human Growth hormone, used in our present work, is a well establish general secretion reporter which has been used on high content screening (Gordon et al. 2016 Traffic) to look for general membrane transport machinery. We have also tested the KDEL function in the context of the immune system analysing the secretion of IL-2. Nevertheless, we take care of this comment and to avoid this conceptual confusion we explained the KDEL biology with more detail in the text.

6- The results shown in Figure 4 are confusing. The inhibition of autophagy using a chemical inhibitor does not prevent lysosomal repositioning. Therefore, the localization of lysosomes and post-Golgi trafficking are not linked to each other. In addition, the basal level of autophagy is pretty high, but the authors state that this is not the case. Nevertheless, the levels of LC3-II in panel b disagree with the text. Furthermore, it seems that it is basal autophagy (i.e. not the starvation induced) that regulates Golgi exit. Conversely, arrival of a cargo wave to the Golgi activates autophagy. Thus, it might be that the high basal levels of autophagy in these cells might be explained by the fact that they have an overloaded ER (ER overload → UPR → upregulation of secretory capacity → more cargo arriving at Golgi → activation of autophagy). This is just one possible scenario, and I am certain more can be thought of. This shows that the level of analysis conducted here requires some further depth.

Our reply: We think the reviewer is referring to the original figure 5e, now figure 6. Indeed, all the other experiments, including the new data, have to be considered to try to understand how several processes converge upon adjusting the Golgi output (secretion) to the Golgi input (ER-to-Golgi trafficking) and cell metabolism (lipid droplet turnover). Taken together all the results point to an scenery where KDEL activation and signalling through PKA stimulates three processes: lysosomal relocation to a perinuclear region, autophagy and lipid droplet degradation. Conditions that perturb any of these processes result in a decreased Golgi-dependent secretion. A secretion reporter becomes accumulated at the Golgi. Lysosomal relocation is not enough to sustain secretion. It has to be associated with the concomitant autophagocytic degradation of LDs. We presented several experiments supporting this conclusion. One of them is mentioned here by the reviewer to conclude that “localization of the lysosomes and post-Golgi trafficking are not linked to each other”. The inhibition of autophagy does not prevent lysosomal repositioning but inhibits secretion. Lysosomal repositioning is not enough. It additionally requires LD degradation by autophagocytosis to support secretion. In addition, we show evidence that lysosome relocation under conditions that enhance autophagy, such as nutrient starvation, is a consequence of autophagosome formation. On the other hand, when we inhibited autophagy (autophagosome formation) by a chemical inhibitor we found that KDEL activation still leads to lysosome repositioning, thus indicating that this relocation process is controlled by KDEL-dependent PKA signalling and not by the autophagic machinery. What remains enigmatic is the requirement of lysosomal perinuclear distribution, close to the Golgi, to sustain post-Golgi trafficking. Nevertheless, we resolved the question of the Reviewers regarding the element that is necessary, in addition to the perinuclear distribution of lysosomes, to promote secretion. The new data point to autophagocytic-dependent LDs degradation as the additional requirement.

We agree respect to the reviewer concern regarding the basal level of autophagocytosis seen in Figure 4. We corrected the text. It is true that HeLa cells transfected to express the secretion reporter have high levels of LC3-II, thus reflecting a high basal level of autophagocytosis. We do not see such levels of LC3-II neither in wild type HeLa cells nor in HeLa cells overexpressing other proteins. As suggested by the Reviewer, an ER overload and the consequently more cargo arriving at the Golgi might activate autophagy. However, the point of the experiment in Figure 4 is still valid. A decreased autophagic activity, down the basal levels, in this case achieved by ATG5 silencing, inhibits secretion. All this is now better explained and commented in the text.

7- In Figure 6, it is necessary to show that p62 is phosphorylated by PKA (using an in vitro kinase assay) and that it is phosphorylated when there is a cargo wave arriving at the Golgi. This can be either done with phosphoproteomics, or by IP of p62 and determining its phosphorylation by immunoblotting.

Our reply: We re-evaluated the putative PKA phosphorylation site on p62 and the new prediction that came out using a different algorithm assigned a lower score for PKA to this site. S182 has been shown to be phosphorylated by other unknown kinase(s). Therefore, in the new version we do not consider the S182 a PKA phosphorylation site. However, the effect of mutating this site to S182 has clear and consistent effects on p62 function. This site belongs to a domain on p62 previously shown to interact with dynein. Our new data show that this site regulates the p62/LD interactions, which is most likely due to interactions with DynLRB1. The expression of p62-S182A mutant reduces the interactions of p62 with LDs, similarly as the DynLRB1-S73A mutant.

8- The role of autophagy is unclear in figures 4-6. It clearly seems that autophagy plays a role in exit from the Golgi. However, instead of providing mechanistic insight, the authors move on and show that selective autophagy (p62-dependent) plays a role. Again it is not clear why type of selective autophagy is involved. Overall, I think that the analysis of the role of autophagy is a bit preliminary and requires more mechanistic depth.

Our reply: We provide new data to support a mechanistic explanation regarding the requirement of ATG5-dependent, p62-dependent autophagy, and lysosome repositioning (DynLRB1-dependent) to sustain cargo secretion. Briefly, lipid droplets (LDs) are reduced by ATG5 silencing, which results in cargo block at the Golgi complex. Restoration of LDs by fatty acid supplementation restored normal cargo transport in these cells. On the other hand, p62 and dynLRB1 regulate LD degradation in a KDEL-dependent manner. The expression of p62 or DynLRB1 mutant blocked the effects of LD restoration in ATG5-silenced cells.

Reviewers' comments:

Reviewer #1 (Remarks to the Author):

The revised manuscript has been improved, however the majority of my concerns has not been included in the revised version (see below). Thus, the manuscript does not fulfil yet the conditions to be published in Nature communication.

- 1) Figures 1b and 2a have been slightly enlarged
- 2) Figure S3a: How long the cells have been treated with the drugs? What is the treatment of the two columns next to those for the control? The authors answer to my question in the rebuttal but this is important to correct the figure legend also.
- 3) Figure 3C: the difference between the two mutants can also reflect a difference in their expression level. It is important to show the quantification of the expression level of the two mutants by western blot
This is still missing in the revised version
- 4) Figure 3D: Lamp1-mCherry is already concentrated near the Golgi in the control cells although it should be dispersed in the cytoplasm when the cargo is in the ER.
This part of the figure has been removed in the revised version. However, given the distribution of Lamp1 in the control cells in this experiment one cannot interpret honestly this experiment.
- 6) Figure legend of 5b-5c is not clear. In the figure legend the authors write that they incubate with the peptide 30 min while they show a time course of 15 min in the graphs. The figure legend should be written according to the figure.

The English has been improved however the figure legend of figure 1a of the revised version as well as its description in the main text are rather difficult to understand.

Reviewer #2 (Remarks to the Author):

The revised manuscript by Tapia et al is much improved. The experimental rationale and logic is much clearer throughout the text, the authors have included more controls and the presentation of the figures is also greatly improved. Overall I think the authors have done a good job at revising the manuscript. I have a few comments which, if addressed would ensure the manuscript reaches the standards and novelty required for Nat. Commun.

I think the autophagy data still raises questions, particularly the role of p62, and requires more work to mechanistically link it with the phenotype the authors are describing. Autophagy clearly plays a role in secretion which is shown in Figure 6. Similarly, Figure 7a shows nicely that p62 is recruited to the Golgi and this is blocked upon inhibition of KDEL signalling. From this point on however, a number of questions are raised. The phosphorylation of p62 has been noted at a number of sites in previous reports, and identification of phosphorylation of S182 comes from a large scale phosphoproteome screen. Furthermore, the interaction of p62 with dynein is not a well-developed function of p62. For these reasons, the author's observations are extremely interesting and important to the field however the authors are attempting to characterise this mutation and its physiological role for the very first time and in Nat. Commun., I believe they need to do this in more detail. The authors say there is no evidence S182 may affect the protein structure but equally, they offer no proof it does not or that it does not affect its function? Does this mutant still bind to LC3 for example? Does it abrogate binding to dynein? Receptor oligomerisation is a main regulatory mechanism for autophagy receptors, is this mutant still able to oligomerise? This mutation is in very close vicinity to the NLS1, does this mutation affect its nuclear shuttling? In fairness, this last comment is probably outside the scope of this study but I believe the point remains that they need to further validate and characterise this mutant which respect to known functions and regulation of p62.

Furthermore, the authors say that p62 S182A reduces formation of autophagosomes, I may have

missed it but I don't see which data indicates that this mutation could affect autophagosome formation per se. You would need to do flux experiments to test this. I also do not think the new data strengthens their conclusion that p62 is acting in its capacity as a selective autophagy receptor rather than in bulk autophagy. Your data indicate that p62S182A is not interacting with lipid droplets properly, but there is no biochemistry to indicate why this may be (such as perturbation of p62-dynein interaction). This may ordinarily might be ok because there is a phenotype except, as I mention above you are characterising the role of this mutation for the very first time.

Minor comments

1. Figure 1c: could you make the red arrows larger?
2. Figure 5b: Could you add to the legend how long the cells were starved for before the peptide was added (i.e. what happened before t=0) because as it is presented now, it takes a few minutes to get your head around exactly how a and b are different.
3. Figure 5b: correct the figure legend? Why does b refer to Dyn-LRB1 transfection? And can you indicate how long the BODIPY-KDEL ligand was incubated with cells for in d-h. Is c in starved or unstarved conditions?
4. Figure 5: I am still unsure why the authors have chosen to show the same assay, i.e. LC3-mCherry-GFP using one method in 5a-c and a different in d-h. It's difficult to easily compare, could the authors at least include an extra panel presenting data in a-c in the same way as I (i.e. autolysosomes/autophagosomes).
5. Figure 8: check spelling SQSMT1 instead of SQSTM1
6. Figure 8i: the western blot looks very different to the quantification
7. Line 308: This sentence doesn't make sense to this reviewer
8. Figure 9b: To me, palmitic acid does rescue secretion as seen by the reduced cargo level on Golgi at 45 and 60 minutes in Atg5 silenced cells. If this is true, the conclusion in line 326-327 does not seem to match.

Reviewer #3 (Remarks to the Author):

The authors have invested some work into revising this manuscript and have dealt with many of my initial remarks. However, the authors were unable to demonstrate that TIDERS is a general response. Thus, I think it is inappropriate to sell the story as such. I also think the arguments why the RUSH did not work are not conclusive. Firstly, the RUSH reporter including its hook remain in the ER and I am not aware of any "leakage" to the Golgi. Therefore it is unclear why the authors observe a lysosomal repositioning after expression of RUSH constructs (which also do not induce ER stress). Moreover, it is not correct that all RUSH systems uses KDEL as retention/hook. Many RUSH cargos exist with Str-li as a retention moiety.

It is likely that the community will not be able to reproduce many of the findings, because most people use the different versions of the RUSH system. My recommendation is therefore:

- down-tone the message and do not oversell the story as a generalized phenomenon
- Show the data demonstrating the problems with the RUSH system. It is important for the community to see these data. This will prevent colleagues from investing time to reproduce the data using the RUSH system and I think this is also in the interest of the authors.

Reviewers' comments:

Reviewer #1 (Remarks to the Author):

The revised manuscript has been improved, however the majority of my concerns has not been included in the revised version (see below). Thus, the manuscript does not fulfil yet the conditions to be published in Nature communication.

Our reply: We thank this Reviewer for pointing out her/his concerns that we have addressed accordingly (see below). We hope that our current revised version of the manuscript has improved in a form that is now suitable for publication in *Nature Communications*.

Reviewer #1:

1) Figures 1b and 2a have been slightly enlarged.

Our reply: Although we are not sure to which Figure 1b this Reviewer is referring, we think it corresponds to former Figures 1c and 2a of the second version of our manuscript. To fulfil as much as possible the request, we have rearranged the layout of Figures 1 and 2. We have removed the graph depicting the quantification of the ratio between lysosomes and Golgi stacks from former Figure 1c. With this extra room, we have enlarged the EM images as much as possible, but to a size permitted by the space constraints. The values of the quantification shown in the removed graph now are depicted in the bottom-left corner of the corresponding EM image. In Figure 2, we moved the EM images from the left side to the top, and now they occupy the entire width of the figure.

Reviewer #1:

2) Figure S3a: How long the cells have been treated with the drugs? What is the treatment of the two columns next to those for the control? The authors answer to my question in the rebuttal, but this is important to correct the figure legend also.

Our reply: Again, although we are not sure to which Figure S3a this Reviewer is referring, we think it corresponds to former Figure S2a of the second version of our manuscript. We pre-treated the cells for 15 min with the drugs, and then maintained the drugs for 30 min in the presence of KDEL-BODIPY. In the graph, the referred two columns next to the control column correspond to treatment with KDEL-BODIPY alone, in the absence of other treatments. To avoid confusion, we changed the figure adding a table below the graph indicating the presence or absence of drugs and KDEL-BODIPY. Also, the figure legend was changed accordingly.

Reviewer #1:

3) Figure 3C: the difference between the two mutants can also reflect a difference in their expression level. It is important to show the quantification of the expression level of the two mutants by western blot.

Our reply: Again, although we are not sure to which Figure 3c this Reviewer is referring, we think it corresponds to former Figure 4c of the second version of our manuscript. As requested, we quantified by western blotting the expression level of the two mutants using an anti-HA antibody, and as loading control anti-actin antibody. The normalized levels of expression of both mutants were similar, and are included in a

bar graph in the second EM image. Representative images of the western blot analysis are also included in the second EM image. All these changes are indicated in the current version of the figure legend.

Reviewer #1:

This is still missing in the revised version

4) Figure 3D: Lamp1-mCherry is already concentrated near the Golgi in the control cells although it should be dispersed in the cytoplasm when the cargo is in the ER.

This part of the figure has been removed in the revised version. However, given the distribution of Lamp1 in the control cells in this experiment one cannot interpret honestly this experiment.

Our reply: Again, although we are not sure to which Figure 3d this Reviewer is referring, we think it corresponds to former Figure 3d of the first version of our manuscript, which we modified in the second version by removing the panels with cell images, leaving only the graph that was identified as Figure 4e. To illustrate better our response, we included below the panels of first version' Figure 3d, but with a different layout. In the figure below, and for explanatory purposes, we also added values of quantifications in the row of images depicting the expression of LAMP1-mCherry. As mentioned above, DynLRB1 expression could not be visualised directly. Therefore, to identify and select cells for live-cell imaging, we co-expressed LAMP1-mCherry and DynLRB1-wt (Control) or DynLRB1-S73A. Under this condition, the distribution of overexpressed LAMP1-mCherry near the perinuclear area is more likely indicative of its biosynthetic traffic from the ER and through the Golgi. On the other hand, when cargo is in the ER, peripheral lysosomes are harder to see because we avoided saturation of the perinuclear fluorescent signal. However, the quantification of the overall distribution of LAMP1-mCherry puncta, assessed by computing the percentage of radial localisation, shows that in control conditions a higher percentage of puncta were in a more perinuclear localisation when cargo reached the Golgi. In contrast, DynLRB1-S73A expression showed no effect in the distribution of LAMP1-Cherry puncta during cargo trafficking. The purpose of removing the panel of images was to avoid confusions or misinterpretations.

Reviewer #1:

6) Figure legend of 5b-5c is not clear. In the figure legend the authors write that they incubate with the peptide 30 min while they show a time course of 15 min in the graphs. The figure legend should be written according to the figure.

Our reply: We thank this reviewer for raising this issue that could lead to misinterpretation. We incubated the cells with the peptides for 15 or 30 minutes, and thus the graphs depict these time periods. We modified the figure legend accordingly, and we think that now is clearer.

Reviewer #1:

The English has been improved however the figure legend of figure 1a of the revised version as well as its description in the main text are rather difficult to understand.

Our reply: We thank this reviewer for this comment that we addressed by modifying the description in the figure legend and the main text.

Reviewer #2 (Remarks to the Author):

The revised manuscript by Tapia et al is much improved. The experimental rationale and logic is much clearer throughout the text, the authors have included more controls and the presentation of the figures is also greatly improved. Overall I think the authors have done a good job at revising the manuscript. I have a few comments which, if addressed would ensure the manuscript reaches the standards and novelty required for Nat. Commun.

Our reply: We thank this Reviewer for her/his encouraging comments. We have performed additional experimental work and modifications to the text to address the majority of her/his concerns and criticisms. We hope this Reviewer will find this revised version improved in a form that now is suitable for publication in *Nature Communications*.

Reviewer #2:

I think the autophagy data still raises questions, particularly the role of p62, and requires more work to mechanistically link it with the phenotype the authors are describing. Autophagy clearly plays a role in secretion which is shown in Figure 6. Similarly, Figure 7a shows nicely that p62 is recruited to the Golgi and this is blocked upon inhibition of KDEL signalling. From this point on however, a number of questions are raised. The phosphorylation of p62 has been noted at a number of sites in previous reports, and identification of phosphorylation of S182 comes from a large scale phosphoproteome screen. Furthermore, the interaction of p62 with dynein is not a well-developed function of p62. For these reasons, the author's observations are extremely interesting and important to the field however the authors are attempting to characterise this mutation and its physiological role for the very first time and in Nat. Commun., I believe they need to do this in more detail. The authors say there is no evidence S182 may affect the protein structure but equally, they offer no proof it does not or that it does not affect its function?

Does this mutant still bind to LC3 for example?

Our reply: We thank this Reviewer for raising this very important issue that we addressed by performing a series of analyses. Because the functional analysis of the effect that the phosphorylation of S182 in p62/SQSTM1 could have in autophagy is beyond the scope of our present study, we toned down the interpretation of the results of the expression of p62/SQSTM1 phospho-mutants as if they represented evidence of an autophagic role. Nevertheless, we evaluated several properties of the p62/SQSTM1 phospho-mutants compared to those of p62/SQSTM1-wt. We first analysed the distribution upon starvation and found that both p62/SQSTM1 phospho-mutants co-localized with LC3-GFP in cytoplasmic puncta that more likely corresponded to autophagosomes. However, the S182A substitution impaired further LC3-GFP degradation, suggesting an important functional role of S182 in autophagy. In addition, we performed co-

immunoprecipitation analyses, which revealed that both p62/SQSTM1 phospho-mutants co-immunoprecipitated with LC3-GFP, indicating that the respective substitutions did not affect this important autophagic function of p62/SQSTM1. These results are described in the main text of the revised manuscript, and are shown in current Supplementary Figure S6.

Reviewer #2:

Does it abrogate binding to dynein?

Our reply: To address this request, we performed a co-immunoprecipitation analysis with samples of cells co-expressing DynLRB1-HA and either of the p62/SQSTM1-mCherry variants. Although the results indicates that each p62/SQSTM1-mCherry variant co-immunoprecipitates with DynLRB1-HA, and thus the S182A substitution did not abrogate the interaction with DynLRB1-HA, we were not able to obtain reproducible results as to whether occurred either reduced or increased level of co-immunoprecipitation, hence this piece of data is not included on our current revised manuscript.

Reviewer #2:

Receptor oligomerisation is a main regulatory mechanism for autophagy receptors, is this mutant still able to oligomerise?

Our reply: We thank this Reviewer for this question that we addressed by performing an oligomerisation assay that is based on the effect of the proteasomal inhibitor MG132. We evaluated the oligomerisation of each p62/SQSTM1-mCherry variant (wt, S182A and S182E) compared to that of endogenous p62/SQSTM1. The results of this analysis are part of current Supplementary Figure S6. We found that all variants were able to oligomerise to an extent that was similar to endogenous p62/SQSTM1, however the S182A mutant showed reduced oligomerisation (see Supplementary Figure S7d). Because oligomerisation might reduce p62/SQSTM1 mobility, we additionally performed FRAP analysis. FRAP showed that all p62/SQSTM1-mCherry variants (wt, S182A and S182E) were able to exchange between cytoplasmic puncta and cytosolic pools, indicating that the respective substitutions did not affect this property. However, the S182A mutant showed increased mobility, whereas the mobility of the S182E mutant was much less (see Supplementary Figure S7). These FRAP analysis are in agreement with the biochemical results, suggesting a functional role of S182 in p62/SQSTM1 oligomerisation and dynamics. These results are described in the main text of our current revised manuscript, and as indicated, are shown in current Supplementary Figures S6 and S7.

Reviewer #2:

This mutation is in very close vicinity to the NLS1, does this mutation affect its nuclear shuttling? In fairness, this last comment is probably outside the scope of this study but I believe the point remains that they need to further validate and characterise this mutant which respect to known functions and regulation of p62.

Our reply: We agree with this Reviewer that providing as much characterisation as possible of the p62/SQSTM1 phospho-mutants is important for better validation. However, as realised by this Reviewer, and as indicated by the journal editor, the analysis of the effect on nuclear shuttling is outside the scope of this study, and thus we have not performed it this time.

Reviewer #2:

Furthermore, the authors say that p62 S182A reduces formation of autophagosomes, I may have missed it but I don't see which data indicates that this mutation could affect autophagosome formation per se. You would need to do flux experiments to test this.

Our reply: We thank this Reviewer for pointing out this issue that indicates we did not state properly the argument, with the potential of causing misunderstanding. We did not mean to state that p62 S182A reduces the formation of autophagosomes. As we mentioned before, we found that upon starvation both p62/SQSTM1 phospho-mutants co-localized with LC3-GFP in cytoplasmic puncta that more likely corresponded to autophagosomes. We also found that the number of LC3-GFP puncta after 30 min of starvation were similar regardless of the p62/SQSTM1 variant expressed, including p62/SQSTM1-wt, indicating that p62 S182A does not reduce the formation of autophagosomes. However, we found that after 60 min of starvation the expression of p62 S182A resulted in accumulation of puncta with increased size containing both LC3-GFP and p62 S182A, suggesting that autophagic flux was impaired. These observations are mentioned in the revised text and shown in current Supplementary Figure 6a.

Reviewer #2:

I also do not think the new data strengthens their conclusion that p62 is acting in its capacity as a selective autophagy receptor rather than in bulk autophagy. Your data indicate that p62S182A is not interacting with lipid droplets properly, but there is no biochemistry to indicate why this may be (such as perturbation of p62-dynein interaction). This may ordinarily might be ok because there is a phenotype except, as I mention above you are characterising the role of this mutation for the very first time.

Our reply:

We thank this Reviewer for raising this issue that we addressed by toning down our conclusions that p62 is acting in its capacity as a selective autophagy receptor. On the other hand, as mentioned by this Reviewer, our data suggest that p62-S182A is not interacting properly with LDs, and we agree in that a biochemical characterisation of this observation is important. However, such biochemical characterisation is beyond the scope of this study. Alternatively, we have performed a quantitative analysis of the observations that we made by live-cell imaging of the interactions between p62 puncta and LDs. We found that the expression of S182A reduced the number of interactions. We also analysed the speed of joint movement and found that the expression of S182A also reduced this. In addition, we analysed the effect of the expression of DynLRB1 phospho-mutants and found that they affected both the number of interactions and the speed of joint movement. These results suggest that S182 is important for the interaction of p62 with LDs, and that this interaction depends on S73 of DynLRB1. We have modified the text to include these new analyses, and also modified Figure 8 that now indicates the values of the quantification.

Minor comments

1. Figure 1c: could you make the red arrows larger?

Our reply: We have increased the arrow size.

2. Figure 5b: Could you add to the legend how long the cells were starved for before the peptide was added (i.e. what happened before t=0) because as it is presented now, it takes a few minutes to get your head around exactly how a and b are different.

Our reply: We included the requested information.

3. Figure 5b: correct the figure legend? Why does b refer to Dyn-LRB1 transfection? And can you indicate how long the BODIPY-KDEL ligand was incubated with cells for in d-h. Is c in starved or unstarved conditions?

Our reply: We have deleted the Dyn-LRB1 referring and included the requested information.

4. Figure 5: I am still unsure why the authors have chosen to show the same assay, i.e. LC3-mCherry-GFP using one method in 5a-c and a different in d-h. It's difficult to easily compare, could the authors at least include an extra panel presenting data in a-c in the same way as I (i.e. autolysosomes/autophagosomes).

Our reply: As requested, for a better comparison with the graph 5i, we have replaced former graph 5c with a graph depicting an autolysosomes/autophagosomes ratio analysis.

5. Figure 8: check spelling SQSMT1 instead of SQSTM1

Our reply: Thanks for detecting this mistake. We corrected the SQSTM1 spelling.

6. Figure 8i: the western blot looks very different to the quantification.

Our reply: As requested, we have replaced the western blot image with a more representative one.

7. Line 308: This sentence doesn't make sense to this reviewer.

Our reply: We agree with this comment and accordingly we removed this sentence.

8. Figure 9b: To me, palmitic acid does rescue secretion as seen by the reduced cargo level on Golgi at 45 and 60 minutes in Atg5 silenced cells. If this is true, the conclusion in line 326-327 does not seem to match.

Our reply: We changed this sentence.

Reviewer #3 (Remarks to the Author):

The authors have invested some work into revising this manuscript and have dealt with many of my initial remarks.

Our reply: We thank this Reviewer for the comment, and we expect that our current revised version is considered improved and suitable for publication in *Nature Communications*.

Reviewer #3

However, the authors were unable to demonstrate that TIDERS is a general response. Thus, I think it is inappropriate to sell the story as such.

Our reply: We think that this comment of this Reviewer could be based on a misunderstanding caused by a not very clear explanation from our part of what we did. Our conclusion that TIDERS is a general response was based on the analyses of the dynamics of lysosomes during the synchronised trafficking of three GFP-tagged cargo proteins, of which two are transmembrane proteins: LAMP1-GFP, Furin-GFP and Albumin-GFP. We also showed the response of an additional previously validated cargo: hGH-GFP-FM. Moreover,

in our current revised version we include the results of three additional cargo proteins in the form of RUSH reporters, further validating the conclusion that TIDERS is a general response (see below).

I also think the arguments why the RUSH did not work are not conclusive. Firstly, the RUSH reporter including its hook remain in the ER and I am not aware of any "leakage" to the Golgi. Therefore it is unclear why the authors observe a lysosomal repositioning after expression of RUSH constructs (which also do not induce ER stress). Moreover, it is not correct that all RUSH systems uses KDEL as retention/hook. Many RUSH cargos exist with Str-li as a retention moiety. It is likely that the community will not be able to reproduce many of the findings, because most people use the different versions of the RUSH system. My recommendation is therefore: - down-tone the message and do not oversell the story as a generalized phenomenon

Our reply: We thank this Reviewer for insisting in the use of RUSH constructs that persuaded us on revising our experimental setup. Of all RUSH constructs that we used, with some experimental modifications that we made, we were able to obtain the expected synchronised behaviour of RUSH-GaIT-GFP, RUSH-EGFR-GFP and RUSH-LAMP1-GFP. In all these three cases we observed lysosome repositioning, validating our conclusion that TIDERS is a generalised phenomenon. We do not know at the moment why the other RUSH constructs that we tried did not behave as expected. More likely it was due to a non-optimal experimental setup from our side. Nevertheless, we estimate that the results using three RUSH constructs are sufficient to add to the conclusion that TIDERS is not dependent on the type of chimeric cargo. On the other hand, we think that this Reviewer comment that **"it is not correct that all RUSH systems uses KDEL as retention/hook"** could be due to a misunderstanding, because we did not mean to say that all RUSH systems use KDEL as retention/hook. In fact, after the experimental problems that we had with the LAMP1-RUSH construct based on the KDEL hook, we used several cargo constructs based on Str-li retention, but also with unexpected results. We tried several modifications to our experimental setup with no success. Again, more likely it was due to a non-optimal experimental setups from our side. Thus, it is likely that the community having more experience with all different RUSH constructs will be able to reproduce the results that we obtained with RUSH-GaIT-GFP, RUSH-EGFR-GFP and RUSH-LAMP1-GFP.

Reviewer #3

- Show the data demonstrating the problems with the RUSH system. It is important for the community to see these data. This will prevent colleagues from investing time to reproduce the data using the RUSH system and I think this is also in the interest of the authors.

Our reply: As we mentioned above, we tried several conditions at using different RUSH constructs during synchronised ER to Golgi transport. We were able to have consistent results only with RUSH-GaIT-GFP, RUSH-EGFR-GFP and RUSH-LAMP1-GFP. The results of these experiments, as well as the respective quantitative analysis, are mentioned in the text of the current revised manuscript and are shown in current Supplementary Figure 1b. The negative results with the other RUSH constructs are however not mentioned in our current revised manuscript, because more likely are the result of suboptimal experimental setup, and thus they could be misleading. In any case, most of the problems seemed to be due to the different expression levels of the RUSH-cargo proteins that we obtained in our transfections. As a general result, we observed that a RUSH-cargo expressed at a relative high level, and regardless of the cell line, required more time to exit the ER, hence affecting the synchronisation. We tried to handle this by incubating with higher concentration of biotin, or longer times of incubation with biotin. Because the optimisation of the assay for

each different RUSH-cargo that we were having troubles was delaying too much our study, we decided to stop and shows the three that worked well.

Finally, we thank the reviewers for their comments, criticism, requests, and corrections that have helped us to strengthen and clarify our manuscript, for which we think is now suitable for publication in *Nature Communications*.

REVIEWERS' COMMENTS:

Reviewer #1 (Remarks to the Author):

My concerns are now included in this revised version and I think that this work can now be published in Nature Communications. Nevertheless, I have still two suggestions:

- 1) Figure 1c I think it is better to add the quantification in the figure legend and in the main text for better readability.
- 2) Figure 4C the graph and the WB for the quantification of the expression level of the mutant, and the Lysosome/Golgi ratio are not easy to read on the EM images. Thus, I suggest to put them below or as a supplementary figure.

Reviewer #2 (Remarks to the Author):

The authors have attempted to address my concerns and have at least characterised the p62 phospho-mutant in more detail. While I still think the manuscript lacks depth, the authors have toned down their conclusions and the further controls and experiments add more robustness to the manuscript. The broad findings of the paper will be of interest to researchers in a number of fields and thus I think the paper warrants publication.

I just have 2 minor points:

- In the text referring to Figure 1, what does $t=0+x(y)(z)$ mean?
- 1c, please give the exact timings for the EM (is it 15 minutes and 45 minutes?)

REVIEWERS' COMMENTS:

Reviewer #1 (Remarks to the Author):

My concerns are now included in this revised version and I think that this work can now be published in Nature Communications. Nevertheless, I have still two suggestions:

1) Figure 1c I think it is better to add the quantification in the figure legend and in the main text for better readability.

Our replay: Quantification in Figure 1c was included in both, main text (lines 110-111) and figure 1c legend as suggested.

2) Figure 4C the graph and the WB for the quantification of the expression level of the mutant, and the Lysosome/Golgi ratio are not easy to read on the EM images. Thus, I suggest to put them below or as a supplementary figure.

Our replay: The graph and WB on Figure 4c were moved below to the EM images as suggested.

Reviewer #2 (Remarks to the Author):

The authors have attempted to address my concerns and have at least characterised the p62 phospho-mutant in more detail. While I still think the manuscript lacks depth, the authors have toned down their conclusions and the further controls and experiments add more robustness to the manuscript. The broad findings of the paper will be of interest to researchers in a number of fields and thus I think the paper warrants publications.

I just have 2 minor points:

- In the text referring to Figure 1, what does $t=0+x(y)(z)$ mean?

Our replay: This part of the text refers to live imaging time course. To avoid confusion, this part of the text was removed from lines 100, 101, 103, 109 and 110. The removed text does not change neither the result nor conclusion of the experiment.

- 1c, please give the exact timings for the EM (is it 15 minutes and 45 minutes?)

Our replay: The time for EM correspond to time 0 (ER) and time 30 minutes (Golgi) are now indicated below each EM image as requested.

Reviewer #3 (Remarks to the Author):

The authors have addressed all my points that I raised. I have no further remarks.